EMBO
Molecular Medicine

# An IgG-based bispecific antibody for improved dual targeting in PSMA-positive cancer

Latifa Zekri[1,2,3,†] (iD), Fabian Vogt[1,†], Lukas Osburg[1], Stefanie Müller[2,3], Joseph Kauer[1,2,3] (iD), Timo Manz[1], Martin Pflügler[1,2,3], Andreas Maurer[4], Jonas S Heitmann[2,3], Ilona Hagelstein[2,3], Melanie Märklin[2,3], Sebastian Hörner[1], Tilmann Todenhöfer[5], Carsten Calaminus[4], Arnulf Stenzl[5], Bernd Pichler[3,4], Christian laFougère[3,6], Marc A Schneider[7], Hans-Georg Rammensee[1,3], Lars Zender[3,8], Bence Sipos[3,8,9], Helmut R Salih[2,3,‡] (iD) & Gundram Jung[1,3,*,‡] (iD)

## Abstract

The prostate-specific membrane antigen (PSMA) has been demonstrated in numerous studies to be expressed specifically on prostate carcinoma cells and on the neovasculature of several other cancer entities. However, the simultaneous expression of PSMA on both, tumor cells as well as tumor vessels remains unclear, even if such "dual" expression would constitute an important asset to facilitate sufficient influx of effector cells to a given tumor site. We report here on the generation of a PSMA antibody, termed 10B3, which exerts superior dual reactivity on sections of prostate carcinoma and squamous cell carcinoma of the lung. 10B3 was used for the construction of T-cell recruiting bispecific PSMAxCD3 antibodies in Fab- and IgG-based formats, designated Fabsc and IgGsc, respectively. In vitro, both molecules exhibited comparable activity. In contrast, only the larger IgGsc molecule induced complete and durable elimination of established tumors in humanized mice due to favorable pharmacokinetic properties. Upon treatment of three patients with metastasized prostate carcinoma with the IgGsc reagent, marked activation of T cells and rapid reduction of elevated PSA levels were observed.

**Keywords** bispecific antibody; immunotherapy; lung cancer; prostate cancer; PSMA

**Subject Categories** Cancer; Immunology

## Introduction

Antibody-based strategies that allow recruitment of T cells against cancer cells have achieved remarkable successes: (i) monoclonal antibodies (mAbs) which block inhibitory "checkpoint" molecules on T cells, such as CTLA4 or PD-1/PD-L1, induce long-lasting remissions even in patients with high tumor burden. However, durable responses are achieved in a minor subset of patients only, and side effects caused by off-target activation of T cells are considerable (Ribas & Wolchok, 2018). (ii) T cells transfected with chimeric receptors (CAR T cells) containing an antibody part directed to the CD19 molecule and CD3-associated signaling domains eradicate high numbers of malignant cells in patients suffering from B-cell-derived malignancies, (Porter et al, 2011; Gross & Eshhar, 2016) and (iii) Blinatumomab, a bispecific antibody (bsAb) with CD19xCD3 specificity in the so-called bispecific T-cell engager (BiTE) format, has received breakthrough designation in acute lymphatic leukemia of B-cell origin (Topp et al, 2011; Riethmuller, 2012; Portell et al, 2013) in 2014.

CAR T cells and T-cell recruiting bsAbs are functionally closely related, and thus, it is not surprising that, besides efficacy, both reagent types share similar toxicity profiles: Despite their tumor-targeting properties, they induce off-tumor T-cell activation, which frequently results in a cytokine release syndrome (CRS) (Maude et al, 2014; Frey & Porter, 2016) that may be successfully treated by

1 Department of Immunology, Institute for Cell Biology, Eberhard Karls University Tuebingen, German Cancer Consortium (DKTK), Partner Site Tuebingen, Tuebingen, Germany
2 Clinical Collaboration Unit Translational Immunology, German Cancer Consortium (DKTK), Department of Internal Medicine, University Hospital Tuebingen, Tuebingen, Germany
3 DFG Cluster of Excellence 2180 "Image-guided and Functional Instructed Tumor Therapy" (IFIT), Eberhard Karls University Tuebingen, Tuebingen, Germany
4 Department for Preclinical Imaging and Radiopharmacy, Werner Siemens Imaging Center, Eberhard Karls University Tuebingen, Tuebingen, Germany
5 Department of Urology, University Hospital Tuebingen, Tuebingen, Germany
6 Department of Nuclear Medicine and Clinical Molecular Imaging, Eberhard Karls University Tuebingen, German Cancer Research Center (DKFZ), Partner Site Tuebingen, Tuebingen, Germany
7 Translational Research Unit, Thorax Clinic at University Hospital Heidelberg, Translational Lung Research Center (TLRC) Heidelberg, Member of the German Center for Lung Research (DZL), Heidelberg, Germany
8 Department of Internal Medicine VIII, University Hospital Tuebingen, Tuebingen, Germany
9 Department of Pathology and Neuropathology, University Hospital Tuebingen, Tuebingen, Germany
*Corresponding author. Tel: +49 7071 29 87621; E-mail: gundram.jung@uni-tuebingen.de
†These authors contributed equally to this work as first authors
‡These authors contributed equally to this work as last authors

steroids or, as approved very recently, the anti-IL6R antibody tocili-zumab (Le *et al*, 2018). In the case of Blinatumomab, CRS limits safely applicable doses to < 30 μg/day, several orders of magnitude lower than the doses applied during therapy with conventional, monospecific antibodies. Upon treatment of solid tumors, a limited access of T cells adds to the dose restrictions imposed by CRS. These problems may be tackled by a careful selection of the target antigen and the format of bispecific antibodies.

Ideally, an optimal target antigen would be expressed not only on tumor cells but also on tumor vessels to allow for sufficient influx of immune cells across damaged vascular endothelium and subsequent tumor cell destruction. PSMA has been reported to be expressed on tumor cells within prostate carcinoma samples as well as on tumor vascular cells in numerous other solid tumors (Chang *et al*, 1999; Buhler *et al*, 2008; Buhler *et al*, 2009). However, questions remain as to simultaneous, dual expression of this antigen.

With respect to format, bsAbs have often been constructed as two covalently linked single chain fragments either in the diabody or BiTE format. The low serum half-life of such molecules ensures good controllability during clinical application, but requires cumbersome continuous infusion regimes (Topp *et al*, 2011). Another disadvantage of these formats is the tendency of many single chain fragments to form aggregates (Worn & Pluckthun, 2001). We feel that this not only constitutes a technical problem for process development, but may cause undesired immunogenicity (Joubert *et al*, 2016) and increase off-tumor T-cell activation. More recently, various IgG-based bispecific formats have been developed. However, several of them retain a tendency to aggregate or lack the stability required for an industrial production process.

Taken together, we intended to overcome the challenges for bsAb development outlined above using the following approaches:

i   Generation and characterization of a new PSMA antibody for improved dual targeting of tumor cells as well as tumor vascular cells.
ii  Development of an improved bsAb format using recombinant constructs with low aggregation tendency and suitable for industrial large-scale production.
iii GMP compliant production and rapid clinical application employing an individualized experimental treatment approach that comprised concomitant medication with the anti-IL-6R antibody tocilizumab to prevent CRS and allow for sufficient dosing.

## Results

### The PSMA antibody 10B3

10B3 is an IgG2b antibody that was obtained using a conventional hybridization procedure as described in the Materials and Methods section. In contrast to J591 (Wang *et al*, 2015), an established PSMA antibody that has been reported to recognize a linear epitope on the native PSMA molecule located at its apical domain (Bander *et al*, 2003b) (Fig 1A), 10B3 is not suitable for the detection of PSMA by Western blot analysis (Fig EV1A). Both antibodies, however, are capable of specifically precipitating PSMA from a prostate carcinoma cell line and, notably, also lung squamous cell

carcinoma (SCC) tumor samples (Fig EV1B and C). This indicates that the 10B3 antibody recognizes a conformational epitope in accordance with (i) the observed lack of cross-inhibition between J591 and 10B3 in FACS-based binding assays (Fig EV1D) and (ii) conjugation mass spectrometry of 10B3-PSMA complexes that revealed four spatially separated stretches of amino acids within the PSMA molecule, 9–14 amino acids in length, that are involved in complex formation (Fig 1A).

Next, 10B3 and J591 were compared by immunohistochemistry on cryosections of various normal and malignant human tissues. Some reactivity of both antibodies was noted with normal prostate epithelium, proximal tubules of the kidney, salivary glands and—to a variable extent—hepatocytes as well as epithelial cells of mammary glands and the gastrointestinal tract, the latter possibly being an artifact caused by excessive mucin production in such tissues. Although the benchmark PSMA antibody J591 and our novel PSMA binder 10B3 showed a similar binding affinity to PSMA *in vitro* (Fig EV1E and F), when we comparatively analyzed their binding in a variety of different solid tumor samples, the results shown in Figs 1B–D and EV2 revealed that (i) in prostate cancer, staining of tumor cells with J591 and 10B3 is comparable, whereas significantly more pronounced binding of 10B3 compared to J591 to the neovasculature was observed in this disease entity (Figs 1B and D, and EV2A); (ii) in SCC samples from a variety of cancer entities other than lung cancer (non-lung SCC, $n = 34$, 28 head and neck cancers, three cervical/uterus cancers, two bladder cancers, one penis carcinoma), very low binding to tumor cells was observed with both antibodies. 10B3 displayed slightly but significantly more pronounced staining. Substantial binding of both J591 and 10B3 to the neovasculature of these non-lung SCC samples was observed, with 10B3 again displaying slightly but significantly higher staining (Fig 1D). This is in line with previously published results reporting on substantial PSMA expression on tumor vessels in various cancer entities (Chang *et al*, 1999); (iii) the most important difference with regard to binding of 10B3 and J591 could be documented in SCC of the lung: here we observed pronounced and significantly higher binding of 10B3 to tumor cells when compared to J591, whereas binding to the neovasculature was rather similar with both antibodies (Fig 1C and D, and EV2B). Of note, vascular and tumor cell staining could be blocked by recombinant PSMA, thereby demonstrating specificity (Fig EV2D). While it cannot fully be excluded that 10B3 might recognize an epitope/protein distinct from PSMA, in our view the immunoprecipitation data shown in Fig EV1A–C (in particular the results shown in Fig. EV1C that were obtained with lung-SCC samples) combined with these blocking experiments clearly show that 10B3 indeed recognizes PSMA. To further validate that binding of 10B3 in lung SCC samples was PSMA specific, we conducted *in situ* hybridization with a PSMA-specific probe (RNAscope) using samples that exhibited variable 10B3 reactivity. These analyses clearly revealed PSMA expression by tumor cells with high and intermediate 10B3 reactivity and in one of two samples with low reactivity (Fig EV3A for illustration). Given a general and pronounced cytoplasmic expression of the PSMA molecule, we further confirmed that 10B3 recognizes membrane PSMA on SCC carcinoma cells by two color immunofluorescence using 10B3 and an antibody directed to the membrane-associated epithelial cell adhesion (EpCAM) molecule (Fig EV3B).

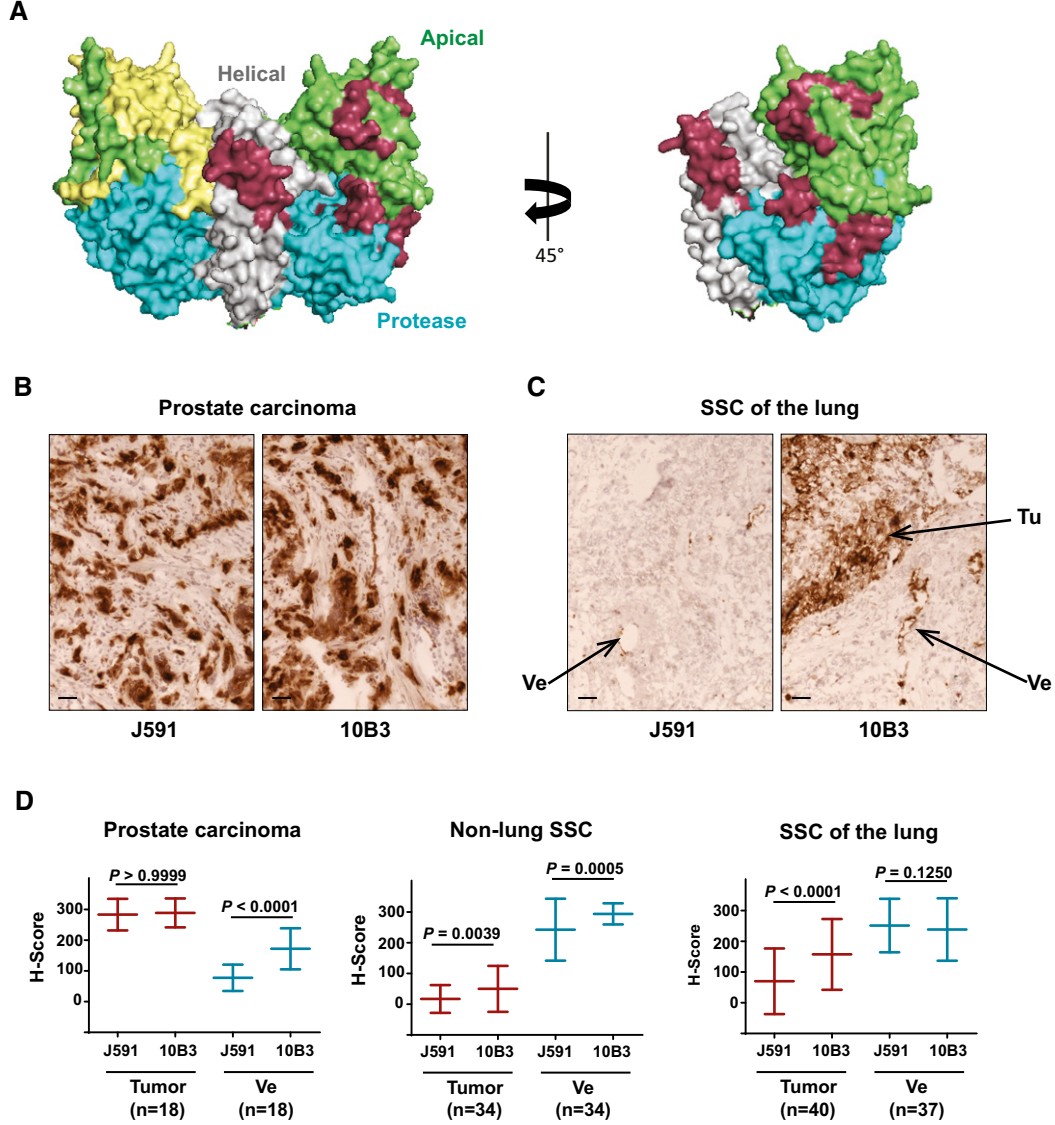

**Figure 1. Binding regions of the PSMA antibodies 10B3 and J591 and comparative reactivity with prostate carcinoma and lung SCC samples.**

A   Binding regions of the 10B3 antibody as determined by conjugation mass spectrometry (in dark red) and of J591 (in yellow) (Bander *et al*, 2003a) are labeled in a 3D model of the PSMA molecule as reported by Davis *et al* (2005) in a dimeric (left panel) or monomeric (right panel) representation of the molecule (PDB ID code 1Z8L). Apical, helical, and protease domains are colored in green, gray, and blue, respectively.

B, C   Cryosections of samples from patients with prostate carcinoma and SCC of the lung. Samples were processed as described in the Materials and Methods section. Ve = vessel, Tu = tumor. Scale 50μm. See also (D) for comparative semi-quantitative analysis of a panel of samples from patients with prostate carcinoma, lung SCC and non-lung SCC.

D   Semi-quantitative analysis of binding of the PSMA antibodies 10B3 and J591 to cryosections from different tumor entities. For definition of the H-score reflecting binding intensity, refer to the Materials and Methods section. Statistical analysis was performed using the paired, non-parametric Wilcoxon test. Sample size (*n*) and the corresponding *P*-values are indicated in the graphs.

## Construction of bispecific PSMAxCD3 antibodies in two different formats

Two different bsAb formats with uni- and bivalent binding to PSMA and CD3, designated Fabsc and IgGsc, respectively, were generated, the latter based on the format originally published by Coloma and Morrison (Coloma & Morrison, 1997). Besides the humanized 10B3 domains, both reagents contain an identical, humanized form of the

CD3 antibody UCHT1 but differ in valency and molecular weight (Fig 2A).

A combination of several point mutations or deletions (Wines *et al*, 2000; Armour *et al*, 2003; Sazinsky *et al*, 2008) was employed to ensure abolishment of Fc receptor (FcR) binding, as this may result in undesired T-cell activation. Fc silencing in CC-1 was confirmed by measuring the binding capacity to recombinant human FcR proteins. In contrast to the corresponding bsAb containing a

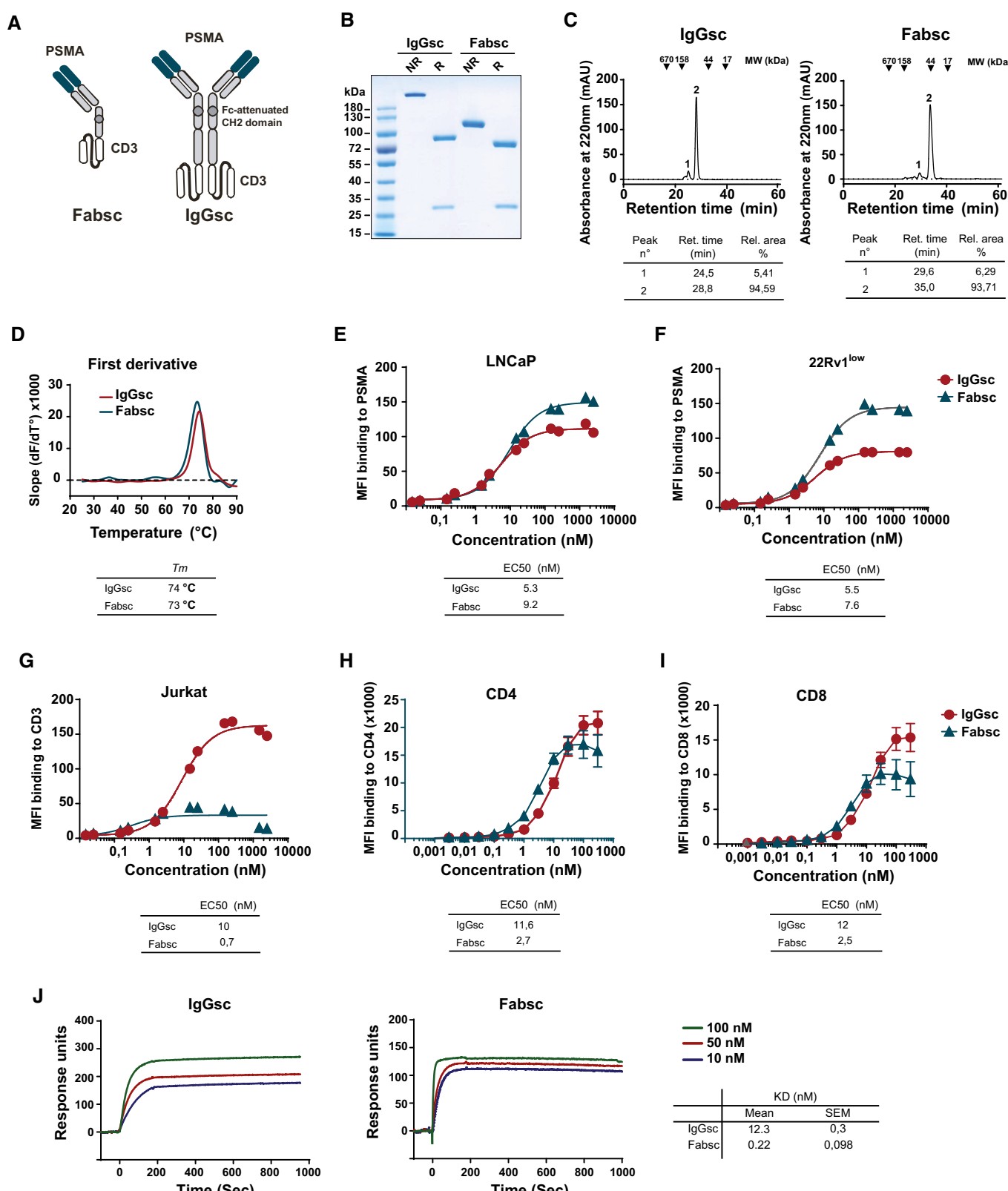

**Figure 2.**

**Figure 2. Characterization of the Fabsc and IgGsc molecules.**

A   Schematic representation of the Fabsc and IgGsc format.
B   SDS–PAGE of IgGsc and Fabsc molecules. NR: non-reduced; R: reduced.
C   Fabsc and IgGsc proteins were subjected to analytical chromatography using Superdex S200 Increase 10/300GL column. Representative gel filtration profiles and the corresponding analysis are presented in the tables below.
D   A fluorescence-based thermal shift assay was performed as described the Materials and Methods section. The data were analyzed using the first derivative approach, and the calculated melting temperatures ($T_m$) are presented in the table below the profiles.
E, F   Binding of the Fabsc and IgGsc molecules to PSMA-expressing LNCaP (E) or 22Rv1$^{low}$ (F) cells was assessed by flow cytometry, as described in the Materials and Methods section. The EC50 values were calculated using GraphPad Prism software and are presented below the figure.
G–I   Binding of the Fabsc and IgGsc molecules to CD3-expressing Jurkat cells (G) and to CD4 T cells (H) and CD8 T cells (I) in PBMC preparations was assessed by flow cytometry. The results represent the average of three independent experiments performed with different PBMC donors. EC50 values are presented in the table below the figure.
J   Binding of the Fabsc and IgGsc molecules to a His-tagged CD3 epsilon-delta heterodimer was determined by SPR. CD3 heterodimer protein was immobilized to an NTA sensor chip and the binding kinetics of the IgGsc or Fabsc molecules was assessed at 25°C. The data obtained were analyzed using Biaevaluation software and the corresponding calculated KD values are presented in the table in the lower right.

wild type Fc part, CC-1 did not bind to any FcR except FcRn (Fig EV4A).

Analysis of the two proteins by SDS–PAGE and gel filtration revealed the expected molecular weights of heavy and light chains and the lack of significant amounts of aggregates, respectively (Fig 2B and C). This was not unexpected, since the CD3 targeting part contained in both molecules was previously selected for minimal aggregation tendency (Durben *et al*, 2015). The stability of both constructs was comparable, as determined by a thermal shift assay (Fig 2D).

Binding of the two bsAbs to PSMA and CD3 expressing cells (LNCaP or 22Rv1$^{low}$ and Jurkat cells, respectively) was assessed by flow cytometry, which revealed EC50 values of approximately 5 and 9 nM for PSMA binding (LNCaP cells) and 10 and 0.7 nM for CD3 binding (Jurkat cells) of the IgGsc and Fabsc molecule, respectively (Fig 2E–G). Whereas the moderate loss of binding affinity of the N-terminal PSMA targeting part of the univalently binding Fabsc molecule was expected, the lower CD3-affinity of the IgGsc molecule that contains two CD3 binding single chain fragments was surprising. The latter was confirmed by measuring the CD3 binding affinities on CD4$^+$ and CD8$^+$ T cells using flow cytometry (Fig 2H and I) and by SPR measurements using recombinant CD3delta epsilon (Fig 2J), respectively. EC50 values for binding to CD3 were comparable between Jurkat and CD4$^+$/CD8$^+$ T cells for the IgGsc molecule, whereas the Fabsc had a lower EC50 activity and a lower plateau level with Jurkat cells as compared to CD4$^+$/CD8$^+$ T cells. Altogether, these findings indicate that: (i) binding is moderately compromised by the bivalent C-terminal arrangement of the two single chains and (ii) the architecture of the TCR/CD3 complex in the membrane of Jurkat cells vs CD4$^+$/CD8$^+$ T cells may differ, with the latter being more accessible to univalent CD3 binders.

## Antitumor activity *in vitro*

For analysis of T-cell activation, tumor cell killing and cytokine release, 22Rv1$^{low}$ and LNCaP cell expressing ~ 3,000 and 40,000

PSMA molecules per cell, respectively, were used as targets (see Materials and Methods section for explanation of PSMA expression during culturing of 22Rv1 cells). In the presence of LNCaP cells, maximal T-cell activation and tumor cell depletion was induced by the two bsAbs at concentrations of ~ 10 pM and at a PBMC:target ratio of 4:1 (Fig 3A and B). With the 22RV1$^{low}$ cells, a higher PBMC ratio (10:1) was required to achieve effective T-cell activation and target cell depletion (Fig 3D and E), whereas EC50 values did not differ substantially and both bsAbs exerted similar activity. Significant differences were noted, however, when the release of cytokines was measured. Levels of IL-6, IL-2, IFN, and TNF (Fig 3C and F) were significantly higher in the presence of the Fabsc compared to the IgGsc molecule, possibly due to the increased affinity of the CD3 binding single chain in the Fabsc format.

Importantly, the IgGsc molecule did not induce any undesired T-cell activation in the absence of target cells, whereas the Fabsc reagent consistently induced unspecific proliferation of T cells at concentrations > 0.1 μg/ml, accompanied by a moderate release of cytokines (Fig 4A and B).

## Pharmacokinetics and antitumor activity *in vivo*

When we determined the serum half-life of the two bsAbs in 57BL/6 mice, we found that the concentration of the Fabsc molecule dropped to 5% of the initial value after 8 h, whereas the half-life of the IgGsc molecule was much longer (Fig 5A). Since this difference was larger than expected when considering the molecular weight of the two molecules (7 vs 16 Ig domains), it appears that the CH3 domains, capable of binding to the FcRn receptor, largely influence the serum half-life of the IgGsc molecule.

Next, we used the chelator p-NCS-Bn-NODAGA to label both antibodies with $^{64}$Cu and demonstrated that the conjugation procedure neither affected binding to PSMA- or CD3-expressing cells nor T-cell activation in the presence of PSMA-expressing target cells (Fig EV4B–D). We then analyzed tumor uptake of the IgGsc

**Figure 3. *In vitro* activity of the Fabsc and IgGsc molecules against tumor cells expressing different amounts of PSMA.**

A–F   BsAb were incubated at the indicated concentrations with PBMC of healthy donors together with LNCaP (A–C) or 22Rv1$^{low}$ cells (D–F) expressing high and low amounts of PSMA, respectively. After 3 days, CD4$^+$ and CD8$^+$ T-cell activation (A, D), target cell depletion (B, E), and cytokine release (C, F) were determined by flow cytometry as described in the Materials and Methods section. Mean values and standard deviations of triplicate measurements are indicated. The dotted line represents the baseline T-cell activation (A, D) or tumor cell counts (B, E) in the absence of bispecific antibodies.

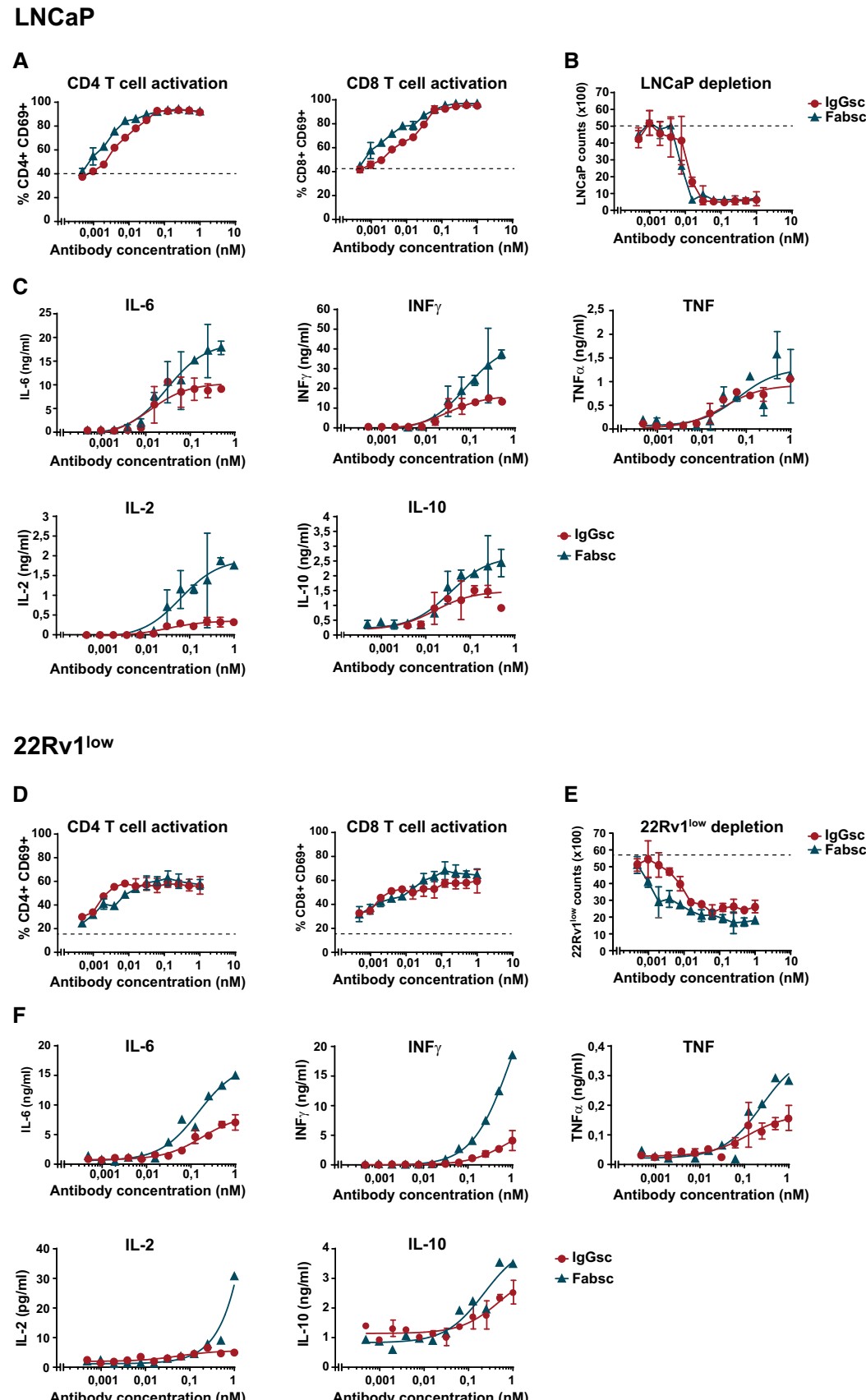

Figure 3.

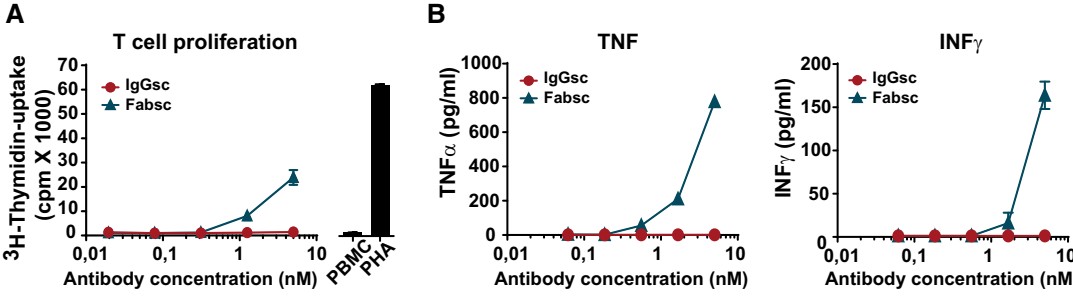

**Figure 4. T-cell activation induced by Fabsc and IgGsc molecules in the absence of target cells.**

A, B PBMC of a healthy donor were incubated with the indicated concentrations of bsAbs, and after 3 days, proliferation (A) and cytokine release (B) were measured using a [3]H-thymidine uptake and a Legendplex assay, respectively. Mean values and standard deviations from triplicate samples.

molecule and found that it localized to PSMA-positive 22Rv1[high] tumors 48 h after injection, but failed to accumulate at PSMA-negative tumor sites (Fig 5B). Thereafter, we determined the uptake of the two different bsAb formats into various tissues of tumor bearing NSG mice over time. Quantitative PET imaging revealed that the Fabsc molecule rapidly localizes to and is secreted by the kidneys; no significant tumor uptake was observed. In marked contrast, lack of renal excretion and a pronounced tumor uptake that increased over time and reached 25% of the applied dose 48 h after injection was observed with the IgGsc molecule (Fig 5C and D). These results were confirmed when the uptake of labeled antibodies to various tissues was determined *ex vivo* after termination of the experiment (Fig 5E and F).

Subsequently, *in vivo* antitumor activity of the constructs was studied using two different mouse models. In a metastasis mouse model, mice were injected with LNCaP cells i.v. (d0) followed by injection of PBMC (10[7]) and antibodies (20 μg) at day 1 and 4. After 21 days, the numbers of metastatic cells in the lungs of the animals were determined by flow cytometry. In this model, the activity of the IgG molecule was clearly superior to that of the smaller molecule, although the Fabsc reagent was injected repeatedly to compensate for its lower serum half-life (Fig 6A).

Next, we employed a second tumor model, in which large tumors were established prior to treatment: LNCaP cells were injected into the right flank of the animals, and treatment was started when tumors had reached a diameter of 5mm. Mice treated with the IgGsc molecule experienced a complete and long-lasting tumor regression at the rather low dose of 2 μg applied three times in weekly intervals together with human PBMC (Fig 6B). In marked contrast, the Fabsc molecule induced only an initial delay of tumor growth, but failed to exert *s*ignificant long-term tumor control, despite the fact that weekly "triple doses" of 2 μg were applied, again to compensate for lower serum half-life. In an attempt to assess the dose dependency of the IgGsc reagent more precisely, we subsequently lowered the applied dose in small increments and found that a dose of 1.2 μg is still effective whereas the slightly lower amount of 1.0 μg induced only a moderate delay of tumor growth and failed to induce durable tumor regression (Fig 6C).

**First in man application**

After completion of GMP production, enabled by public funding from the Helmholtz Association, the IgGsc protein, designated CC-1,

was applied to three patients with metastasized, prostate carcinoma refractory to standard therapy. Before treatment, all patients underwent MRT and PET scans with a PSMA-specific tracer that demonstrated PSMA expression at multiple tumor sites (Fig EV5A). CC-1 was then applied in the course of an individualized experimental treatment approach using a daily dose escalation, from 20–40 μg up to 2.6 mg. To prevent a clinically relevant cytokine release syndrome (CRS), the anti-IL-6R antibody Tocilizumab was applied as soon as a fever of ≥ 38.5° occurred for the first time. In contrast to i.v. application of steroids that are widely used to prevent CRS, tocilizumab does interfere with T-cell activation neither *in vitro* nor *in vivo* (Kauer *et al*, 2020).

Under these conditions, serum concentrations for CC-1 of up to 200–500 ng/ml were reached, which is more than two orders of magnitude higher than those achieved with the benchmark bsAb Blinatumomab. Patient 1 experienced no relevant side effects, patient three suffered from an urticaria that subsided after topical application of steroids. Patient 2 developed a pulmonary edema that ultimately required high dose i.v. steroid treatment for full recovery. Notably, retrospective analysis revealed that this patient, who had a history of allogeneic stem cell transplantation for a hematopoietic malignancy, had pre-existing human anti-human antibodies that may have enhanced the elimination of CC-1, but importantly also the protective antibody tocilizumab. In all patients, profound T-cell activation and a rapid and marked decline of PSA levels were observed, which rose again 20–30 days after cessation of CC-1 treatment (Figs 7 and EV5B).

# Discussion

Since the late nineties, it is well known that PSMA antibodies not only bind to tumor cells within prostate carcinoma samples, but also to the neovasculature of a variety of other solid tumors, e.g., breast and colon carcinoma, melanoma, and glioblastoma (Haffner *et al*, 2012; Ren *et al*, 2014; Wernicke *et al*, 2014). However, questions remain regarding the expression of PSMA on (i) tumor cells in non-prostate malignancies and (ii) the neovasculature specifically in prostate carcinoma. With respect to the first, Mhawech-Fauceglia and coworkers reported PSMA expression for 154 of 2,174 investigated tumor samples of non-prostate origin, among them 26 of 297 positive SCC samples (Mhawech-Fauceglia *et al*, 2007). More recent publications

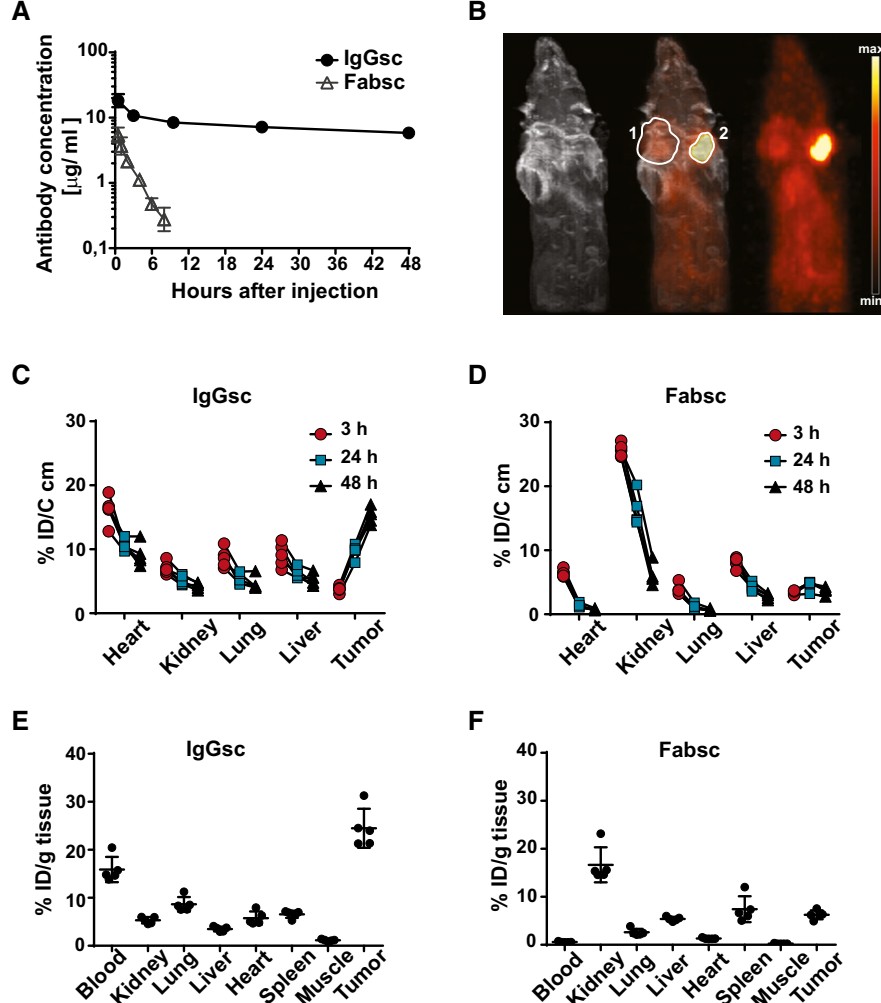

**Figure 5. Half-life and tumor localization of the Fabsc and the IgGsc molecules in immunodeficient mice.**

A 20 µg of the indicated bsAbs was injected i.v. into 57BL/6 mice, and serum concentrations were measured at the indicated time points after injection using the Promega assay described in the Materials and Methods section. Mean values and standard deviations obtained from groups of four animals per time point are indicated.

B [64]Cu labeled bsAbs were injected into SCID mice carrying established DU145 (PSMA-negative, marked by "1") and 22Rv1[high] (PSMA-positive, marked by "2") tumors on opposite flanks. MRT and PET scans were obtained after 48 h.

C–F [64]Cu labeled bsAbs were injected into NSG mice carrying 22Rv1[high] tumors (five animals per group), and uptake at the indicated locations was determined by PET quantification over time (C, D) and at termination of the experiment after 48 h (E, F) as described in the Materials and Methods section. Error bars represent standard deviation of the mean value.

reported an at least weak PSMA expression on tumor cells in pancreatic carcinoma and again in SCC of the lung (124/147 and 47/87 positive cases, respectively) (Ren *et al*, 2014; Wang *et al*, 2015).

Our study confirms PSMA expression on lung SCC tumor cells that was particularly pronounced if the novel 10B3 antibody was used. 10B3 staining of tumor cells was detected by conventional immunohistology as well as RNA *in situ* hybridization. The staining was specific as it was blocked by recombinant PSMA. It is not exclusively cytoplasmic but includes a membrane expression as demonstrated by double staining with an antibody directed against the EpCAM molecule.

Concerning vascular expression in prostate carcinoma, it was reported in early seminal papers that it is not detectable in most samples, which is in stark contrast to findings reported with

multiple other solid tumor entities (Silver *et al*, 1997; Chang *et al*, 1999). However, in some of these studies PSMA antibodies that bind to denatured PSMA were used, whereas the two PSMA antibodies compared in our study bind to PSMA in its native form (Tykvart *et al*, 2014). In our experiments, PSMA expression on the neovasculature of prostate carcinoma was less pronounced than that observed in SCC specimen, but nevertheless clearly detectable in most samples, in particular if 10B3 was used for staining. Taken together, binding of 10B3 is markedly superior to that of J591, with regard to both, neovascular cells in prostate carcinoma and tumor cells in lung SCC samples. This may be due to the peculiar conformational epitope recognized by 10B3. Analysis of binding by both, surface plasmon resonance and flow cytometry binding assays

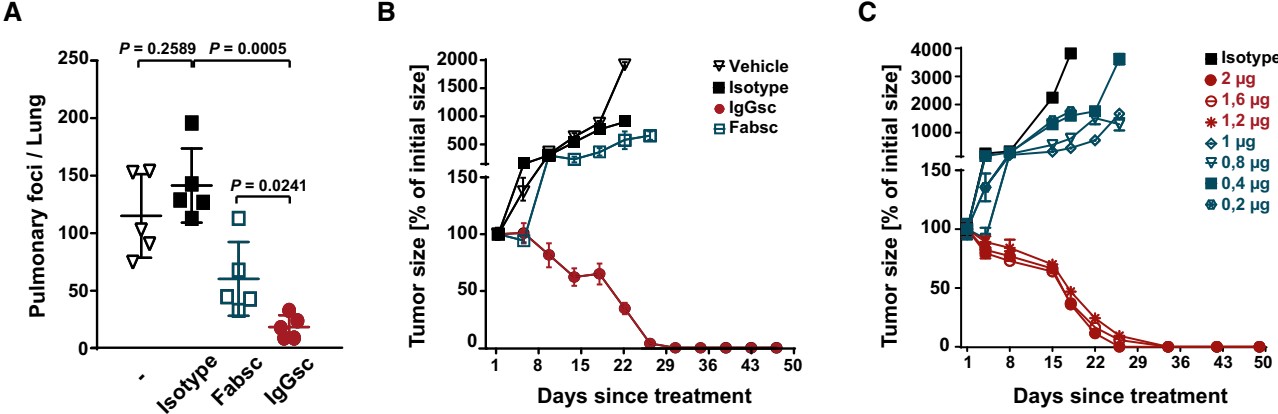

**Figure 6. Antitumor activity of the Fabsc and IgGsc molecules in immunodeficient mice.**

A    $4 \times 10^6$ LNCaP cells were injected into NSG mice (five per group) followed 1 day later by transfer of human PBMC ($10^7$ i.p.) and application of bsAbs (10 μg) at day 1 and 4 (IgGsc) and daily at day 1–6 (Fabsc), respectively. At day 21, mice were killed and tumor cells in the lung were quantified by flow cytometry after enzymatic digestion. Statistical analysis was performed using the unpaired t-test. Error bars represent standard deviation of the mean value.

B, C    $10^6$ LNCaP cells were injected s.c. into the right flank of NSG mice (five per group), and treatment with PBMC and bsAbs was started when tumors had reached a diameter of 5 mm. BsAbs (2 μg per dose) were applied at day 1, 8, and 15 (IgGsc) and daily at day 1–3, day 8–10, and day 15–17 (Fabsc), PBMC was applied at day 1, 8, and 15. Mean values and standard deviations of five mice per group are indicated.

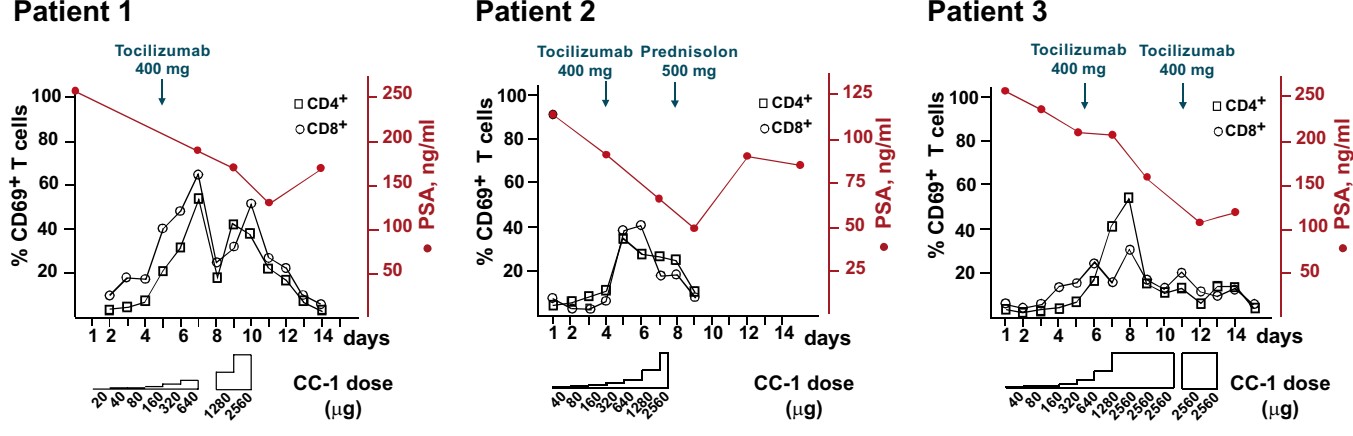

**Figure 7. T-cell activation and PSA levels during treatment of three patients with castrate resistant, metastasized prostate carcinoma.**

The IgGsc molecule was applied in escalating doses, with a starting dose of 20 μg (patient 1) or 40 μg (patients 2 and 3) to a maximum of ~ 2.6 mg, as indicated. PSA values were monitored and T-cell activation was assessed by flow cytometry.

showed comparable affinities for J591 and 10B3 (Fig EV1E and F). This indicates that the differences observed by immunohistology might be attributable to a better accessibility of the 10B3 epitope once the cells are organized within a tissue. With respect to the conformational nature of this epitope, conjugation mass spectrometry has revealed 4 stretches of amino acids within the PSMA molecule that are involved in conjugate formation with the 10B3 antibody. Whereas simultaneous binding of the antibody to all four stretches may appear unlikely, an extensive analysis of numerous previously analyzed 3D structures of antigen antibody complexes has revealed that contact regions of antibodies directed to native proteins are relatively large and may easily reach the size of a protein domain (Stave & Lindpaintner, 2013). Alternatively, binding of 10B3 could induce a conformational change of the PSMA molecule that affects the homogeneity of the population of molecules during crosslinking mass spectrometry procedure. In any case, based on our findings, 10B3 appears particularly suited for dual targeting of tumor cells and associated tumor vessels in prostate carcinoma as well as lung SCC. Dual targeting may allow for sufficient influx of immunological effector cells to the tumor site across a damaged or inflamed endothelial barrier in line with reports demonstrating that even large numbers of tumor-specific T cells fail to exert sufficient antitumor activity unless a proinflammatory environment has been generated at the tumor site. This may be critical

for sustained therapeutic success of T-cell-based therapies in general and of bsAb treatment in particular (Sondel *et al*, 1989; Ganss *et al*, 2002).

Besides the choice of a suitable targeting antibody, the selection of a stable format is another critical issue in bsAb development. In general, a recent tendency in the field that also applies to bsAb targeting PSMA (Buhler *et al*, 2008; Buhler *et al*, 2009; Fortmuller *et al*, 2011; Friedrich *et al*, 2012; Baum *et al*, 2013; Hernandez-Hoyos *et al*, 2016; Patterson *et al*, 2017) favors IgG-based molecules rather than smaller proteins in the BiTE or diabody format. However, so far it is not clear, whether those molecules perform better with respect to stability and function, since data providing side by side comparison of "small" and "large" bispecific molecules are scarce. When we compared the antitumor activity of our "small" Fabsc and the "large" IgGsc molecule, only moderate differences with regard to antitumor activity were observed *in vitro*. However, unwanted off-target T-cell activation and cytokine release were somewhat higher with the smaller Fabsc molecule, probably due to the higher affinity of its CD3 binding moiety. Likewise, in the absence of target cells the Fabsc molecule induced a moderate activation of T cells, while the IgGsc reagent was inactive. This implies that bivalent CD3 stimulation by the IgGsc molecule does not result in undesired off-target T-cell activation or cytokine release.

Compared to the *in vitro* experiments, the differences between the two molecules noted after *in vivo* application were much larger: First, the radiolabeled Fabsc molecule failed to localize at PSMA-expressing tumors due to rapid renal excretion, which occurred despite a molecular weight of ~ 80 kDa that exceeds the albumin threshold. In marked contrast, serum half-life of the IgGsc molecule was much longer, and a significant portion of the molecule was found at the tumor 48 h after injection.

In both, a metastasis prevention model and in a second model where mice were bearing large established tumors, the IgGsc molecule achieved a marked and prolonged antitumor effect, whereas the Fabsc bsAb was clearly less efficient and ineffective, respectively. Notably, this holds true despite the application schedule accounted for the lower serum half-life of the Fabsc bsAb. These results support the notion that the maintenance of sufficient serum levels is of critical importance for the therapeutic activity of bsAbs. The activity of the IgGsc molecule against established tumors was maintained at weekly doses as low as 1.2 µg, corresponding to a dose of ~ 4 mg in humans. We certainly share the view that testing bsAb in immunodeficient mice adoptively transferred with human immune cells has only limited value for the prediction of therapeutic activity in humans; Nevertheless, these data may serve to guide initial clinical development of CC-1 at least to some extend.

Due to increasing regulatory requirements, successful completion of a translational process is a formidable challenge for academic institutions (Hemminki & Kellokumpu-Lehtinen, 2006; Salih & Jung, 2019) Supported by the Helmholtz validation fund, the GMP compliant production of CC-1 was recently completed and a first in man study enrolling patients with castrate-resistant prostate carcinoma (CRPC) has meanwhile been initiated (ClinicalTrials.gov.NCT04104 607). The design of this study differs from the protocols used for clinical evaluation of the benchmark bsAb Blinatumomab in as much as it aims to achieve serum concentrations significantly exceeding 1 ng/ml to achieve optimal therapeutic activity. To this end, tocilizumab rather than steroids is used as concomitant medication to prevent CRS because this anti-IL-6R antibody—in contrast to steroids—does not interfere with the antitumor activity of CC-1 (Kauer *et al*, 2020).

The application of CC-1 to three patients within individualized treatment approaches was based on the considerations outlined above. Although it is obviously too early to draw definite conclusions, the achieved serum levels and the rapid and marked PSA reduction observed during treatment of these patients indicate that tocilizumab may be suited to effectively attenuate the sequelae of cytokine release, thereby allowing for dosing of CC-1 that results in substantial serum levels. The presently ongoing clinical study will ultimately reveal whether the rapid decline of PSA values observed in all three patients reported in this work indeed translates to a sustained clinical benefit.

# Materials and Methods

### Cells and reagents

PBMCs were isolated from heparinized blood of healthy donors by density-gradient centrifugation (Biocoll separating solution, Biochrom, Berlin, Germany). The prostate carcinoma cells lines 22Rv1, LNCaP and DU145 as well as HEK293 cells were purchased from the German collection of microorganisms and cell cultures (DSMZ, Braunschweig, Germany), the Jurkat-cell line from the American Type Culture Collection (ATCC, Manassas, USA). All cell lines listed above are of human origin and were cultured as described previously (Hofmann *et al*, 2012; Durben *et al*, 2015). The Chinese hamster ovary derived FreeStyle™ CHO-S cells were cultivated in FreeStyle™ CHO Expression Medium (Life Technologies, Darmstadt, Germany), supplemented with 8 mM L-glutamine (Lonza) at 37°C and 7.5% $CO_2$. All cell lines were regularly tested for mycoplasma contamination.

Expression of PSMA on 22Rv1- and LNCaP cells was determined by flow cytometry using QIFIKIT calibration beads according to the instructions of the manufacturer (Agilent, Santa Clara, USA). 22Rv1 cells were initially found to express ~ 20,000 molecules per cell (22Rv1[high]) and were used for *in vivo* imaging studies. Upon long-term culture, PSMA expression declined to a stable value of 3,000 molecules per cell (22Rv1[low]). Those cells were used in addition to LNCaP cells, that stably expressed high PSMA levels (~ 40,000 molecules per cell) as targets for *in vitro* activity assays.

### 10B3 hybridoma generation, sequencing, and humanization

The IgG2b antibody 10B3 was generated after immunization of female BALB/c mice with irradiated Sp2/0-Ag14 cells (ATCC) transfected with human PSMA. Variable regions of heavy (VDJ) and light (VJ) chains were sequenced by Aldevron GmbH (Freiburg, Germany).

The $V_H$ and $V_L$ sequences were searched against the human germline sequence databases with IgBLAST http://www.ncbi.nlm. nih.gov/igblast/) and IMGT/V-QUEST (http://www.imgt.org/IMGT_vquest/input). The residues within KABAT CDR regions were grafted onto the framework regions of human κ light chain sequence IGKV3-20*02 and of the heavy chain sequence IGHV3-11*06. The T20 humanness score as defined by Gao *et al* (2013) rose from 75.5 to 83.7 and from 66.5 to 84.0 for the VH and VL domain, respectively.

The affinity of the humanized molecule was ~ 5 nM and comparable to that of the murine parental version.

## Co-immunoprecipitation assays and Western blotting

22Rv1, HEK-293 cells (20 × 10$^6$ cells), and slices of Lung squamous cell carcinoma tumors (10 × 10 μm) were lysed and immunoprecipitated as described previously (Zekri *et al*, 2013). Briefly, precipitated proteins were eluted with sample buffer from protein G agarose beads (Roche) and separated by SDS–PAGE. Western blotting was performed by incubating the membranes with the indicated primary murine anti-PSMA antibodies (1:500) or anti-tubulin antibody (1:5,000, Tub 2.1 clone; Sigma) for 1 h followed by incubation with secondary anti-Mouse IgG (H + L), HRP conjugate (1:10,000 Promega) and detection by the AceGlow chemiluminescent substrate (Peqlab), as recommended by the manufacturer.

## Crosslinking conjugation mass spectrometry

Crosslinking mass spectrometry was performed by CovalX AG, Zürich, Switzerland. Briefly, non-covalent conjugates between recombinant PSMA, purchased from Sino Biochemicals (Bejing, China, Cat.Nr. 15877-H07H), and purified 10B3 antibody were stabilized by incubation with a specifically developed crosslinking mixture (Bich *et al*, 2010). The covalent binding generated allows the complexes to survive the sample preparation process and thus mass spectrometry analysis by a special high-mass detection system.

## Immmunohistochemistry, H-score, and RNA *in situ* Hybridization

Cryosections of different human tissues were purchased as tissue microarrays (FDA Standard frozen tissue array) from BioChain (Newark, USA). Squamous cell carcinoma samples of different origin as well as prostate carcinoma sections were provided by the Departments of Pathology and Urology, respectively, of the University of Tübingen as well as by the lung biobank of the German Cancer Research Center, Heidelberg. Sections were fixed with acetone and stained with primary murine J591, 10B3, and CD31 antibodies (10 μg/ml) followed by ZytoChem Plus (HRP) Polymer anti-Mouse kit from Zytomed Systems GmbH, (Berlin, Germany). In the blocking experiment presented in Fig EV2D, directly consecutive tumor sections were stained in the absence or presence of recombinant PSMA protein (50 μg/ml). The slides stained with biotinylated CC-1 antibodies (5 μg/ml) were initially blocked with avidin-biotin blocking kit and subsequently detected by a streptavidin-HRP conjugate (Zytomed Systems GmbH, Berlin, Germany).

The results were evaluated by a semi-quantitative approach assigning an H-score (or "histo" score) to different tumor samples. Staining index of 0–3 to each sample corresponding to absent (0), weak (1), intermediate (2), and strong (3) staining defined as containing no, 1–10, 11–50, and >50% of specifically stained cells, respectively. Evaluation of samples was performed by an experienced pathologist (BS). For each histological entity, an H-score was defined (ranging from 0–300) as sum of respective staining indices ×100 divided by the number of samples.

RNA *in situ* Hybridization was performed using the RNAscope® assay, developed by Advanced Cell Diagnostics (Hayward, CA). To this end, 27 ZZ probes were designed to target the Hs-FOLH1 mRNA nucleotide sequences [85–397] and [2,666–3,960] (GenBank no. NM_004476.3), using the RNAscope® Probe Design tool. Probes targeting the housekeeping gene UBC and the bacterial gene *dapB* were used as positive and negative controls, respectively. Hybridization was performed on 10 μm cryosections of PSMA-positive lung squamous cell carcinoma tumors using RNAscope®2,5 HD reagent kit brown-Hs according to the instructions of the manufacturer (Advanced Cell Diagnostics, Hayward, CA).

For detection of PSMA on tumor cell membranes by two color immunofluorescence, cryosections of lung SCC samples were stained with 10B3 and Cy3-labeled secondary goat anti-mouse secondary antibody (Jackson/Dianova) followed by an antibody directed to the human EpCAM molecule (clone 215) that was conjugated to FITC using fluorescein-isothiocyanate (Sigma, St. Louis, USA).

## Generation, purification, and characterization of recombinant bispecific antibodies

Variable domain sequences of 10B3 antibody, J591 antibody (anti-PSMA; GenBank no. FR853148.1 and FR853149.1), MOPC-21 (used as a control antibody; GenBank no. AAD15290.1 and AAA39002.1), and humanized UCHT1 (anti-CD3; GenBank no. AAB24133.1 and AAB24132.1) were codon optimized (*Homo sapiens*) using the GeneArt GeneOptimizer tool (Thermo Fisher Scientific). $V_H$, $V_L$, and scFv sequences were synthesized *de novo* at GeneArt (Thermo Fisher Scientific, Regensburg, Germany). The variable sequences were inserted into a human Igγ1 backbone comprising $C_H1$-$C_H2$-$C_H3$- or $C_K$-constant domain sequences as described previously (Hofmann *et al*, 2012). For the generation of bsAbs, a scFv fragment of the humanized UCHT1 ($V_L$-$V_H$) was inserted at the C-terminus of the Fab or IgG sequence of 10B3 and MOPC-21. UCHT1 $V_L$ and $V_H$ were joined by a flexible (GGGGS)$_3$ linker. The IgGsc construct is based on the format originally described by Coloma and Morrison (Coloma & Morrison, 1997) with modifications in the $C_H2$ domain consisting of the amino acid substitutions and deletions E233P; L234V; L235A; ΔG236; D265G; A327Q; A330S (EU index) which abrogate FcR binding and complement fixation. The Fabsc format was constructed as previously described (Durben *et al*, 2015). Fabsc and IgGsc bsAbs were purified from culture supernatants by affinity chromatography on KappaSelect and Mabselect affinity columns, respectively. The antibodies were subjected to analytical and preparative size exclusion chromatography using Superdex S200 Increase 10/300GL and HiLoad 16/60 columns (GE Healthcare), respectively, and only the fractions containing the monomeric form were used. Endotoxin levels of samples as determined by a limulus amebocyte lysate assay (Endosafe®Charles River, Charleston, SC) were < 0.5 EU/ml.

For some experiments, sequences coding for the murine variable regions of 10B3, J591, and MOPC-21 were fused to human constant regions to generate chimeric versions of the respective monospecific antibodies, designated as ch10B3, chJ591, and chMOPC-21, respectively.

## Kinetic, stability analysis, and ELISA assay

Surface Plasmon Resonance (SPR) experiments with recombinant, His-tagged PSMA produced by transfected CHO cells were performed using 10B3 and J591 antibodies immobilized to a sensor Chip coated with Protein A (GE Healthcare, Chicago, USA). Binding

was analyzed using a Biacore X instrument at 25°C at a flow rate of 30 μl/min. Binding to CD3 was assessed using an NTA sensor and reagent kit for capture of Histidine-tagged molecules (GE Healthcare, Chicago, USA). To this end, His-tagged CD3 epsilon-delta heterodimer protein (Acro Biosystems) was immobilized to the sensor chip and the binding kinetics of the IgGsc or Fabsc molecules were recorded at a flow rate of 10 μl/min.

To further test protein stability, a fluorescence-based thermal shift assay was performed using the Proteostat® thermal shift stability assay kit (ENZO Life Sciences, New York, USA) according to the manufacturer's instructions. Thermal denaturation was carried out by increasing the temperature from 25 to 95°C at a rate of 1°C per minute. The data were analyzed using the first derivative approach to calculate the corresponding melting temperature ($T_m$).

FcR binding analysis was conducted using ELISA by coating wells his-tagged FcγRI, FcγRIIb, FcγRIIa, or FcRn protein (R&D Systems, Minneapolis, MN, USA) Then, bsAbs were added to the plate at the indicated concentrations, and binding was visualized using an HRP-conjugated goat anti-human-Fc antibody (Jackson ImmunoResearch, West Grove, PA, USA). Unless indicated, all experiments were performed at a neutral pH.

### In vitro activity of bispecific antibodies

BsAb-induced T-cell activation, target cell depletion, and cytokine release were determined using flow cytometry. To this end, PBMCs from healthy donors and bsAbs were incubated with LNCaP or 22Rv1[low] cells at the indicated effector to target ratios in a 96-well plate. Activated lymphocytes were detected using CD4-PacificBlue (clone RPA-T4), CD8-FITC (clone HIT8a), and CD69-PE (clone FN-50). Target cells, identified as CD14- CD276+, were stained with CD14-APC/Cy7 (clone HCD14) and CD276-PE/Cy7 (clone MIH42). Viability was accessed using 7-AAD. All directly labeled antibodies and the corresponding isotype controls were from BioLegend (San Diego, CA). Absolute cell numbers were calculated using equal numbers of BD™CompBead (BD Biosciences). For detection of primary antibodies, PE-conjugated goat anti-human F(ab)2 fragments were used (Jackson ImmunoResearch, West Grove, PA, USA). For multiplex cytokine analysis, supernatants were harvested after 24 or 72 h of incubation. Cytokines were measured using the Th1 LEGENDplex™ multiplex kits (BioLegend) according to the manufacturer's instructions.

In some experiments, T-cell activation in the presence or absence of irradiated tumor target cells was assessed using a ³H thymidine uptake assay as described previously (Durben et al, 2015).

### Radiolabeling of antibodies and in vivo imaging, mouse models

Conjugation of antibodies to the chelator p-NCS-Bn-NODAGA (Chematech, Dijon, France) and preparation of the ⁶⁴Cu in a PET trace cyclotron (GE Healthcare, Uppsala, Sweden) were performed as described previously (Rolle et al, 2016). Radiochemical purity and antibody integrity were analyzed by TLC on silica plates and by size exclusion chromatography, respectively. Radiochemical purity of the protein lots used for imaging was 96.0–96.5%.

Animal experiments were carried out with the approval by the local animal care committee at the University of Tübingen and in accordance with German federal and state regulations.

**The paper explained**

**Problem**

Bispecific antibodies (bsAbs) directed against a tumor-associated target antigen and the T-cell receptor (TCR)/CD3 complex can re-direct T cells against cancer cells, sometimes resulting in complete remissions, as observed in patients with B-cell-derived leukemia. The activity of bsAb against solid tumors, however, has been less impressive, probably due to the limited accessibility of such tumors to T cells. For this reason, an ideal target antigen would be expressed not only on tumor cells but also on tumor vessels to allow sufficient influx of immune cells across a damaged vasculature (dual targeting).

**Results**

The prostate-specific membrane antigen (PSMA) is expressed on prostate carcinoma cells, as well as on vascular cells within many solid tumor types. We have generated a PSMA antibody that recognizes a conformational epitope and appears to be particularly suited for dual targeting. We have used this antibody to construct bsAbs in two different formats based on Fab and IgG scaffolds (Fabsc and IgGsc). The antitumor activity of both molecules in vitro was comparable. In vivo, IgGsc efficiency was superior due to its longer half-life and better tumor localization.

**Impact**

We have developed an IgG-derived bispecific antibody for improved dual targeting of PSMA-expressing tumors. GMP compliant production has been completed and a first-in-man study in patients with prostate cancer is ongoing (NCT04104607).

For the dual-xenograft models, $4 \times 10^6$ 22Rv1[high]- and DU145 cells were injected subcutaneously at both shoulders of female SCID mice. For the single xenograft model, 22Rv1[high] were inoculated at the right flank of female NSG mice. Imaging studies were performed when the tumors had reached an estimated size of 500 mm³. The radiolabeled protein ($10 \pm 1.3$ MBq) was injected through a tail vein catheter, and positron emission tomography (PET) scans and subsequent magnetic resonance imaging (MRI) were performed 3, 24, and 48 h after injection. Briefly, PET images were reconstructed using an ordered subset expectation maximization (OSEM-2D) algorithm. Reconstructed PET and MR images were manually fused, volumes of interest were contoured based on MR information, and tracer accumulation was analyzed at an Inveon Research Workplace (Siemens Preclinical Solutions). Results are expressed as percentage of the applied dose per cubic centimeter (%ID/cc). After the final imaging session, biodistribution was analyzed ex vivo using a Wizard2 gamma counter (Perkin Elmer, Waltham, MA, USA). Decay-corrected radioactivity was normalized for organ weight, and results are expressed as %ID/g.

For determination of serum half-life, C57BL/6 mice were injected i.v. with 20 μg of Fabsc and IgG antibodies and serum concentrations of the proteins were determined using a bioluminescent cell-based assay system developed by Promega (Madison, USA).

For the metastasis model, $4 \times 10^6$ LNCaP cells were injected into NSG mice i.v. (day 0) followed 24 h later (day 1) by human PBMC ($10^7$ cells, i.p.). IgGsc and Fabsc antibodies (10 μg i.v.) were applied at day 1 and 4 (IgGsc) and daily at day 1–6 (Fabsc), respectively. After 21 days, the number of tumor cells in the lung was determined by flow cytometry after enzymatic digestion of the removed organ.

Cells were stained with antibodies directed to mCD45, huCD45, and CD276, the latter serving a specific marker for LNCaP cells, and were then analyzed by flow cytometry.

To evaluate the activity of the bispecific molecules against established tumors, $10^6$ LNCaP cells were injected s.c. into the right flank of female NSG mice and treatment was started when tumors had reached a diameter of 5 mm (day 1). PBMC ($10^7$ i.p.) and IgGsc antibodies (2 µg i.v.) were applied at day 1, 8, and 15, Fabsc antibodies were applied at the same doses daily at day 1–3, day 8–10, and day 15–17. Tumor growth was measured twice weekly, and mice were euthanized if tumors had reached a diameter of 15 mm or at signs of graft vs host disease (GvHD).

**Clinical application of CC-1**

Three patients with relapsed, metastasized, and castrate-resistant prostate carcinoma were treated within an individualized experimental therapy regime, pursuant to section 13 of the German Medicines Act (AMG) and in accordance with the principles of the WMA Declaration of Helsinki (section 37) and the department of health and human services Belmont report. The treatment regime is based on a clinical trial that presently is recruiting (NCT 04104607). All participants were extensively informed about the nature of the treatment and provided written consent. Besides daily standard clinical and laboratory assessment, patients were monitored daily for T-cell activation by flow cytometry using the activation marker CD69.

## Data availability

This study includes no data deposited in external repositories.

**Expanded View** for this article is available online.

### Acknowledgements

We acknowledge the expert technical assistance by Carolin Walker, Beate Pömmerl, Dennis Thiele, and Christine Beschorner. We also appreciate the support by the Lung Biobank Heidelberg, a member of the Biomaterial bank Heidelberg (BMBH), and the Biobank Platform of the German Center for Lung Research (DZL). In addition, this work was supported by the German Cancer Consortium, the Helmholtz validation fund (OPTIMAB), an innovation grant administered through the University of Tübingen to FV as well as grants from the Deutsche Forschungsgemeinschaft (SA1360/9-1 and SA1360/7-3), Wilhelm Sander-Stiftung (2007.115.3), Deutsche Krebshilfe (111828 and 70112914), and Germany's Excellence Strategy (EXC 2180/1).

### Author contributions

FV generated and characterized the 10B3 antibody and performed its humanization together with TM. LatZ conducted most *in vitro* and *ex vivo* experiments presented in the manuscript with the help of LO. JK organized and follow-up all the immunohistochemistry with the help of SH. SM and MM performed the animal experiments under the supervision of HRS. IH performed the assay for the detection of bsAbs in mouse serum. MP organized the GMP compliant production. JSH was substantially involved in monitoring experiments during clinical application with the help of JK. AM and CC labeled the bsAb and performed the imaging studies under the supervision of BP. TT, AS, and LarZ contributed substantially to clinical care for the patients. BS evaluated the immunohistochemistry data. ClF conducted patient PET scans. MAS provided the SSC lung tumor samples. H-GR was conceptually involved in interpretation of the data and the design of functional *in vitro* assays. GJ designed and supervised most functional *in vitro* and *in vivo* experiments, the latter together with HRS, and wrote the paper together with HRS and LatZ.

### Conflict of interest

GJ, HS, FV, and LatZ are listed as inventors on the patent application "Novel PSMA binding antibody and uses thereof," EP16151281, applicant German Cancer Research Center, Heidelberg, Germany.

### For more information

https://clinicaltrials.gov/ct2/show/NCT04104607
https://www.krebsdaten.de/Krebs/EN/Content/Cancer_sites/Lung_cancer/lung_cancer_node.html
https://www.proteinatlas.org/ENSG00000086205-FOLH1
https://www.medizin.uni-tuebingen.de/en-de/das-klinikum/einrichtungen/kliniken/medizinische-klinik/kke-translationale-immunologie/ag-salih
https://www.immunology-tuebingen.de/groups/gundram-jung.html

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
