## [Review Process File · EMBO Molecular Medicine]

An IgG-based bispecific antibody for improved dual targeting in PSMA positive cancer

Latifa Zekri, Gundram Jung, Helmut Salih, Fabian Vogt, Lukas Osburg, Stefanie Müller, Joseph Kauer, Timo Manz, Martin Pflügler, Andreas Maurer, Jonas Heitmann, Ilona Hagelstein, Melanie Märklin, Sebastian Hörner, Tilmann Todenhöfer, Carsten Calaminus, Arnulf Stenzl, Bernd Pichler, Christian la Fougère, Marc Schneider, Hans-Georg Rammensee, Lars Zender, and Bence Sipos
DOI: 10.15252/emmm.201911902

Corresponding authors: Gundram Jung (gundram.jung@uni-tuebingen.de) , Latifa Zekri (l.zekri-metref@dkfz-heidelberg.de)

Review Timeline:

Submission Date:	13th Dec 19
Editorial Decision:	12th Feb 20
Revision Received:	26th May 20
Editorial Decision:	6th Jul 20
Revision Received:	5th Oct 20
Editorial Decision:	20th Oct 20
Revision Received:	26th Nov 20
Accepted:	1st Dec 20

Editor: Lise Roth

Transaction Report:

12th Feb 2020

Dear Gundram,

Thank you for the submission of your manuscript to EMBO Molecular Medicine. Please accept my apologies for the delay in getting back to you, which is due to the fact that one referee did not return his/her report despite several reminders. In order to avoid delaying the process further, we prefer to make a decision now based on the two reports we received.

As you will see, while both referees mention the potential translational interest of the study, they also raise substantial concerns on your work, which should be convincingly addressed in a major revision of the present manuscript. In particular, both referees noted a lack of rigor in several places, such as in the characterization of the antibody, the choice of in vivo models, and the unclear correlation between preclinical and human studies.

Addressing the reviewers concerns in full (above points as well as other reviewers' comments) will be necessary for further considering the manuscript in our journal. Still, revising the manuscript according to the referees' recommendations appears to require a lot of additional work and experimentation, and I am unsure whether you will be able or willing to address those and return a revised manuscript within the 3 months deadline. On the other hand, given the potential interest of the findings, I would be willing to consider a revised manuscript with the understanding that acceptance of the manuscript would entail a second round of review. EMBO Molecular Medicine encourages a single round of revision only and therefore, acceptance or rejection of the manuscript will depend on the completeness of your responses included in the next, final version of the manuscript. Should you find that the requested revisions are not feasible within the constraints outlined here and prefer, therefore, to submit your paper elsewhere, we would welcome a message to this effect.

When submitting your revised manuscript, please carefully review the instructions that follow below. Failure to include requested items will delay the evaluation of your revision:

- 1) A .docx formatted version of the manuscript text (including legends for main figures, EV figures and tables). Please make sure that the changes are highlighted to be clearly visible.
- 2) Individual production quality figure files as .eps, .tif, .jpg (one file per figure).
- 3) A .docx formatted letter INCLUDING the reviewers' reports and your detailed point-by-point responses to their comments. As part of the EMBO Press transparent editorial process, the point-by-point response is part of the Review Process File (RPF), which will be published alongside your paper.
- 4) A complete author checklist, which you can download from our author guidelines (<https://www.embopress.org/page/journal/17574684/authorguide#submissionofrevisions>). Please insert information in the checklist that is also reflected in the manuscript. The completed author checklist will also be part of the RPF.
- 5) Please note that all corresponding authors are required to supply an ORCID ID for their name

upon submission of a revised manuscript.

6) Before submitting your revision, primary datasets produced in this study need to be deposited in an appropriate public database (see <https://www.embopress.org/page/journal/17574684/authorguide#dataavailability>). Please remember to provide a reviewer password if the datasets are not yet public. The accession numbers and database should be listed in a formal "Data Availability " section (placed after Materials & Method). Please note that the Data Availability Section is restricted to new primary data that are part of this study.

7) We would also encourage you to include the source data for figure panels that show essential data. Numerical data should be provided as individual .xls or .csv files (including a tab describing the data). For blots or microscopy, uncropped images should be submitted (using a zip archive if multiple images need to be supplied for one panel). Additional information on source data and instruction on how to label the files are available at .

8) Our journal encourages inclusion of *data citations in the reference list* to directly cite datasets that were re-used and obtained from public databases. Data citations in the article text are distinct from normal bibliographical citations and should directly link to the database records from which the data can be accessed. In the main text, data citations are formatted as follows: "Data ref: Smith et al, 2001" or "Data ref: NCBI Sequence Read Archive PRJNA342805, 2017". In the Reference list, data citations must be labeled with "[DATASET]". A data reference must provide the database name, accession number/identifiers and a resolvable link to the landing page from which the data can be accessed at the end of the reference. Further instructions are available at .

9) We replaced Supplementary Information with Expanded View (EV) Figures and Tables that are collapsible/expandable online. A maximum of 5 EV Figures can be typeset. EV Figures should be cited as 'Figure EV1, Figure EV2" etc... in the text and their respective legends should be included in the main text after the legends of regular figures.

- Additional Tables/Datasets should be labeled and referred to as Table EV1, Dataset EV1, etc. Legends have to be provided in a separate tab in case of .xls files. Alternatively, the legend can be supplied as a separate text file (README) and zipped together with the Table/Dataset file. See detailed instructions here:

10) The paper explained: EMBO Molecular Medicine articles are accompanied by a summary of the articles to emphasize the major findings in the paper and their medical implications for the non-specialist reader. Please provide a draft summary of your article highlighting

11) For more information: There is space at the end of each article to list relevant web links for further consultation by our readers. Could you identify some relevant ones and provide such information as well? Some examples are patient associations, relevant databases, OMIM/proteins/genes links, author's websites, etc...

12) Every published paper now includes a 'Synopsis' to further enhance discoverability. Synopses are displayed on the journal webpage and are freely accessible to all readers. They include a short stand first (maximum of 300 characters, including space) as well as 2-5 one-sentences bullet points that summarizes the paper. Please write the bullet points to summarize the key NEW findings. They should be designed to be complementary to the abstract - i.e. not repeat the same text. We encourage inclusion of key acronyms and quantitative information (maximum of 30 words / bullet point). Please use the passive voice. Please attach these in a separate file or send them by email, we will incorporate them accordingly.

Please also suggest a striking image or visual abstract to illustrate your article. If you do please provide a jpeg file 550 px-wide x 400-px high.

13) As part of the EMBO Publications transparent editorial process initiative (see our Editorial at <http://embomolmed.embopress.org/content/2/9/329>), EMBO Molecular Medicine will publish online a Review Process File (RPF) to accompany accepted manuscripts.

In the event of acceptance, this file will be published in conjunction with your paper and will include the anonymous referee reports, your point-by-point response and all pertinent correspondence relating to the manuscript. Let us know whether you agree with the publication of the RPF and as here, if you want to remove or not any figures from it prior to publication.

I look forward to receiving your revised manuscript.

With my best wishes,

Lise

Lise Roth, PhD
Editor
EMBO Molecular Medicine

To submit your manuscript, please follow this link:

Link Not Available

Photos 400-800 DPI

Figures are not edited by the production team. All lettering should be the same size and style; figure panels should be indicated by capital letters (A, B, C etc). Gridlines are not allowed except for log plots.

Figures should be numbered in the order of their appearance in the text with Arabic numerals. Each Figure must have a separate legend and a caption is needed for each panel.

***** Reviewer's comments *****

Referee #1 (Comments on Novelty/Model System for Author):

please see critique. figures lack rigor in presentation so it is unclear if results are internally consistent. the table demonstrating prevalence and intensity of staining is confusing. the murine model is not ideal.

Referee #1 (Remarks for Author):

The authors generate a novel PSMA-specific antibody and describe its novelty in targeting both tumor and vascular components; they further construct a bsAb which they evaluate in a xenograft model as well as a pilot clinical trial for toxicity.

There is certainly significant interest in developing optimal bsAbs that are both effective and safe, and the authors claim that their PSMA bsAb addresses both these challenges.

However, there are several major flaws in their studies that undermine their conclusions and the manuscript, as presented, does not provide a coherent portrayal of the significance and broader applicability of their findings.

Major criticisms

1. The authors claim that 10B3 exhibits dual targeting properties as it recognizes a conformational state of PSMA and is thus able to target tumor vascular endothelial as well as PSMA+ tumor cells

(in contrast to the prototypic anti-PSMA antibody, J591 which only recognizes tumor cells). They provide an IHC of SSC of lung (fig 1 c) and demonstrate 10B3 staining of presumably vascular endothelial cells. In the same slide staining is shown for 10B3 staining of SSC lung tumor cells.

- a. No counter staining of Ve is performed confirming these are Ve cells (e.g CD105 a marker of tumor vasculature, or any other Ve-specific marker)
 - b. Absence of J591 staining in SSC is claimed to be the result of inferior recognition of PSMA by this antibody vs 10B3. The authors do not perform any other studies on the same slide demonstrating that staining of Tumor (and Ve) by 10B3 is the result of PSMA recognition. Fig EV4 is cited as evidence that 10B3 stains PSMA+ tumor cells by RNAscope, yet there is no indication that this is the same section as that presented in Fig 1c and no expression of PSMA in Ve is demonstrated here. That this is not presented in a clear and unequivocal fashion is concerning given that this is one of the major findings of the manuscript
2. The authors then proceed to use 10B3 as both Fabsc and IgGsc constructs. However, there is no demonstration that either form recognizes Ve cells.
 3. A first-in-man clinical study is then performed. Understandably, the hurdles that must be overcome to move this into the clinic can be formidable and appear to have been addressed for the trial to be realized. However, adding this to this manuscript, due to lack of meaningful correlates, gives the impression that this was 'tacked on'. Expression of PSMA in the tumors and tumor vasculature of these patients is not presented. The PSA results presented do not give any idea of the velocity of PSA prior to treatment or extended results beyond the relatively short interval presented. It is unclear if PSA was already falling or the difference seen is due to fluctuations within a relatively narrow range of values. The toxicity mitigation with tocilizumab is not significant given the small number of patients nor surprising given its use in other CAR-T cell therapies.

Minor criticisms

1. The wording in several key sentences is awkward; several run-on sentences make reading difficult (e.g. second para, page 7 "For comparative analysis..."). Redundant verb use is unnecessary (e.g. last sentence 3rd para, page 6). These are only some examples. A native English speaker to review the manuscript for readability and clarity is recommended.
2. Fig 5 c - appears to be mislabeled - tumor eradication is seen at doses of 0.8, 1.0 and 2.0 but not at doses of 1.6, or 1.2 which is both internally inconsistent and contrary to the text in manuscript.
3. Table 1 is very difficult to read and understand. Perhaps an H-score to summarize both intensity and % cells + would have been more useful

Referee #3 (Comments on Novelty/Model System for Author):

Zekri et al claim to have made a novel antibody with improved dual-targeting capabilities, however it is unclear what the impact of this is. Their model system primarily looks at PSMA(+) tumor cells, and aside from IHC does not discuss the PSMA(+) vasculature enough. Although their antibody may indeed be a more clinically suitable therapeutic, the models used did not reflect this adequately.

Referee #3 (Remarks for Author):

Zekri et al identified and engineered a bispecific antibody for the treatment of PSMA(+) cancer with T-cells. Patients with prostate cancer have limited treatment options and any novel approaches that limit toxicity while shrinking tumor are likely to have a major impact the field. In this manuscript, Zekri et al cover two important points: First they describe how this anti-PSMA clone differs from the conventionally used J591 clone, by showing how the epitope of 10B3 is more widely displayed in neovasculature of multiple cancer types. Second they engineer two T-cell bispecific antibodies

using this sequence and evaluate their function in vivo, both pre-clinically and in three patients, providing first in human data for this T-cell bispecific antibody. Although two BsAb formats were compared, it is unclear why either was chosen among the big list of possible antibody platforms. The superiority of 10B3 over J591 was not adequately explained or evident in the data. Overall the manuscript provides interesting data regarding PSMA targeting and BsAb formats. With revisions it should be suitable for consideration.

The following changes are recommended to more clearly articulate the functional relevance of 10B3 compared to J591, in vitro:

- 1) Fc-attenuation is mentioned but not clearly cited or spelled out in the manuscript. Which mutations were used to ablate Fc binding? How did this attenuation affect antigen binding, pharmacokinetics, and T cell homing?
- 2) Reformat Table 1 to display the relative fraction of samples that stain at each intensity level. This can be both in a table, or even a graph, such that the differences between tumor types can be more clearly seen. In addition, statistical analysis between relevant groups should be presented to prove the novel reactivity with neovasculature.
- 3) Also reformat Table 1 such that the frequency of staining within each intensity level can more clearly be compared between J591 and 10B3. For example, splitting each tumor type into a 2-cell wide by 4-cell long table for each tumor type and antigen type (Tumor, Vasculature) would allow one to compare J591 and 10B3 side by side for each intensity level. A similar change can be made for the aggregate staining scores to allow for statistical comparison.
- 4) It is not clear if these scores and the slides they were based on were evaluated by a trained pathologist. Although a pathologist is included in the author list. If he or she has evaluated these slides, it would be important to highlight this in the methods or even the text to strengthen the claims.
- 5) Figure EV3 shows very little difference between 10B3 and J591, relative to the slides shown in figure 1. Please comment on how representative these slides are of their respective staining intensity as displayed in table 1, and if necessary, provide additional examples of when these antibodies stain similarly or differently.
- 6) Figure EV4B does not clearly show the experimental intent of the authors. More robust quantification or single color controls or control tissues would help.
- 7) Figure 4B needs to have the legend updated to address more clearly what 1 and 2 are labeling.

The following changes are recommended more clearly explain the current in vitro and in vivo BsAb data, and more robustly evaluate its potential in other indications:

- 1) The figure legend of Figure EV5 C-E seems to be inconsistent with the text. In the figure, CD3 binding is clearly superior for the IgGsc, while PSMA binding is moderately improved with the Fabsc, in contrast to the text, which describes it in the opposite way. Please either correct or clarify. This is also seen on page EV7, which shows the same results as EV5.
- 2) Additionally, it would be useful to evaluate the CD3 binding to primary human PBMCs or T-cells, in addition to or instead of Jurkat cells.
- 3) The results of Figure EV5C-E are somewhat surprising either way. Could this be from the staining method used? The anti-Fab2 may bind differently to the Fabsc compared to the IgGsc. For example, could the IgG-sc molecule be bound by two different secondary molecules? Additional controls to evaluate this would be valuable. Additionally, while gel-filtration data looks clear, numerical summary of each peak for the SEC-HPLC would be helpful. If these values differ substantially it may be worth re-testing the staining using purer preparations, i.e. without aggregates.
- 4) Why does Fabsc show consistently better IFN and IL-2 release in vitro? This suggests that Fabsc may be aggregating in vitro, or may be contaminated in some way (endotoxin, etc). Please

provide data supporting that Fabsc and IgG-sc have similar in vitro stabilities and endotoxin levels. Assuming they are the same, please provide a better explanation of why the Fab sc seems to perform so well. This would also impact why Fab-sc functions independent of tumor.

5) The binding data is referred to as affinity, which is slightly misleading. Given that PSMA binding kinetics were evaluated by SPR, it would be very helpful to both present this data earlier in the manuscript, (along with the other biochemical characterizations) and include the fitted affinity quantitation's. This should be done for both BsAb's and using CD3 as well. Doing so will allow for more appropriate conclusions to be made about binding affinities and binding kinetics (for comparison between different formats and antibody clones).

6) The LNCAP data appears compelling, but it would be helpful to compare this against a J591 BsAb as well. Although this is not entirely necessary in the case of LNCAP, it would be useful to show that 10B3 can indeed control or shrink non-prostate PSMA(+) tumors in vivo. It is understood that this level of biology may not be possible to model using a cell line or PDX xenograft system, but an explanation of such limits would be helpful too. In the absence of in vivo data, a clearer explanation of why the authors believed 10B3 to be superior to J591 would be helpful. For example, does the reduced specificity of 10B3 warrant concern that 10B3 would be more toxic to normal vasculature? Could it also reduce the tumor targeting of the antibody, and therefore reduce T-cell infiltration?

7) More details are needed in the method section regarding the outline of the animal models used. Routes of administration, timing and doses need to be more clearly spelled out. In addition, justification is needed for starting treatment so soon after tumor implantation (24hr, page 8). Please also update the figure axis labels to distinguish days since treatment or days since implantation.

8) On page 9, the following sentence is unclear: "In our hands, this antibody, in marked contrast to steroids, interferes with the anti-tumor activity of CC-1 neither in vitro nor in vivo". Please rephrase or clarify.

9) It was unclear how the clinical trial was designed. The exact regulatory authority who approved the study (NCT 04104607) should be spelt out. For example, what was the justification for the dosing schedule and regimen. Additional, clearer quantitation of the CC-1 doses in Figure 6 would be helpful. In vitro cytokine release started at 0.01 nM (~2 ng/ml), peaking at 1 nM (~200 ng/ml) of BsAb. Serum concentrations of 200-500 ng/ml should hit the peak of cytokine storm - a dosing regimen seemingly incompatible with safe design in a phase I bispecific antibody trial. A serum level of 200-500 ng/ml for any bispecific antibody (whether BiTE or IgG formats) is very high and potentially lethal. It is possible that this construct was not very effective in activating T cells. The mention of steroids affecting anti-tumor activity should be referenced not as unpublished results but with citation from previous publications.

10) Regarding patient 2's response to tocilizumab, it is explained that they may have had a pre-existing anti-human immune response. If so, shouldn't this have also impacted the level of CC-1 in the blood, not just Toci?

Manuscript number: EMM-2019-11902

Referee #1 (Comments on Novelty/Model System for Author):

Referee#1 (Remarks for Author):

The authors generate a novel PSMA-specific antibody and describe its novelty in targeting both tumor and vascular components; they further construct a bsAb which they evaluate in a xenograft model as well as a pilot clinical trial for toxicity.

There is certainly significant interest in developing optimal bsAbs that are both effective and safe, and the authors claim that their PSMA bsAb addresses both these challenges.

However, there are several major flaws in their studies that undermine their conclusions and the manuscript, as presented, does not provide a coherent portrayal of the significance and broader applicability of their findings.

Major criticisms

1. The authors claim that 10B3 exhibits dual targeting properties as it recognizes a conformational state of PSMA and is thus able to target tumor vascular endothelial as well as PSMA+ tumor cells (in contrast to the prototypic anti-PSMA antibody, J591 which only recognizes tumor cells). They provide an IHC of SSC of lung (fig 1 c) and demonstrate 10B3 staining of presumably vascular endothelial cells. In the same slide staining is shown for 10B3 staining of SSC lung tumor cells.

a. No counter staining of Ve is performed confirming these are Ve cells (e.g CD105 a marker of tumor vasculature, or any other Ve-specific marker)

We thank the reviewer for this remark. To address this issue we obtained series of 3- μ m cryo-slides of lung SCC samples and stained them with 10B3, J591 or anti-CD31 (used as an established vascular marker). To facilitate evaluation of vascular staining, we have selected a section with a predominantly vascular expression of PSMA. The new Figure EV2B demonstrates that 10B3 and -to a somewhat lesser extent- J591 resemble the vascular staining pattern obtained with the anti-CD31 antibody. Slides were evaluated and vascular expression patterns were confirmed by an experienced pathologist listed as coauthor (BS).

b. Absence of J591 staining in SSC is claimed to be the result of inferior recognition of PSMA by this antibody vs 10B3. The authors do not perform any other studies on the same slide demonstrating that staining of Tumor (and Ve) by 10B3 is the result of PSMA recognition. Fig EV4 is cited as evidence that 10B3 stains PSMA+ tumor cells by RNAscope, yet there is no indication that this is the same section as that presented in Fig 1c and no expression of

PSMA in Ve is demonstrated here. That this is not presented in a clear and unequivocal fashion is concerning given that this is one of the major findings of the manuscript

We thank the reviewer for rising this point. To further address the specificity of 10B3 staining, a blocking IHC experiment was conducted. Briefly, series of consecutive 3- μ m lung SSC cryo-sections were obtained and each slide stained with 10B3 antibody in the presence or absence of 50- μ g of recombinant PSMA protein. The results are included in the new Figure EV2D. The experiment showed that staining with 10B3 can be blocked by an excess of recombinant PSMA suggesting that it is PSMA specific.

2. The authors then proceed to use 10B3 as both Fabsc and IgGsc constructs. However, there is no demonstration that either form recognizes Ve cells.

We thank the reviewer for this remark and would like to point out that we have previously performed an extensive immunohistological characterization of biotinylated CC-1 in the IgGsc format. Respective results are part of the documents submitted to the regulatory authority in charge for approval of a clinical study. As expected, the observed staining pattern of CC-1 was very similar, if compared to that observed for the parental 10B3 antibody. With the peculiarity that T cells were additionally stained by the bsAb due to its CD3 binding. To address the reviewer concern completely we have performed an additional immunohistological analysis of biotinylated CC-1 in the IgGsc and Fabsc format using lung SCC samples, and an exemplary staining is presented in the new figure EV2C. It can be seen that the staining pattern achieved with these molecules are similar and resemble the vascular staining pattern observed with the anti-CD3 antibody. Again it should be noted that in the case of the bsAb, additional staining of lymphocytes may occur. Given this peculiarity one can safely conclude that the staining of SCC samples by both bsAbs was very similar to that observed for the parental monospecific PSMA antibodies.

3. A first-in-man clinical study is then performed. Understandably, the hurdles that must be overcome to move this into the clinic can be formidable and appear to have been addressed for the trial to be realized. However, adding this to this manuscript, due to lack of meaningful correlates, gives the impression that this was 'tacked on'.

Expression of PSMA in the tumors and tumor vasculature of these patients is not presented.

We thank the reviewer for raising this important issue and now provide additional information on the PSMA/PET scans of the three patients prior to CC-1 treatment. Respective images

are included in the revised version, figure EV5A and clearly show for all patients, localization of a PSMA specific tracer to multiple tumor sites.

The PSA results presented do not give any idea of the velocity of PSA prior to treatment or extended results beyond the relatively short interval presented. It is unclear if PSA was already falling or the difference seen is due to fluctuations within a relatively narrow range of values.

We appreciate this valuable comment and have included additional information on the PSA values of the three patients before, during and after CC-1 treatment. This information is presented in Figure EV5B and confirms that, indeed, a rapid decline of PSA values was observed during CC-1 application, followed by a moderate increase after cessation of treatment.

The toxicity mitigation with tocilizumab is not significant given the small number of patients nor surprising given its use in other CAR-T cell therapies.

We certainly agree that data obtained from a limited number of patients are to be interpreted with due care. We also agree that the therapeutic activity of Tocilizumab for treatment of established CRS is well documented. In our paper, this antibody was used in a prophylactic setting. In a recently accepted manuscript from our group (Kauer et al.2020, in press), we have demonstrated that early Tocilizumab application prevents bsAb mediated CRS by maintaining a normal body temperature and relatively low CRP (C-reactive protein) levels at rather high serum concentrations of IL-6. If requested we could provide this manuscript.

Minor criticisms

1. The wording in several key sentences is awkward; several run-on sentences make reading difficult (e.g. second para, page 7 "For comparative analysis..."). Redundant verb use is unnecessary (e.g. last sentence 3rd para, page 6). These are only some examples. A native English speaker to review the manuscript for readability and clarity is recommended.

To address this issue of the reviewer, we have rephrased the paragraphs mentioned above in the revised manuscript as following:

Paragraph page 7: "For analysis of T-cell activation, tumor cell killing and cytokine release, 22RV1^{low} and LNCaP-cells expressing approx. 3,000 and 40,000 PSMA molecules per cell,

respectively, were used as targets (see materials and methods section for explanation of PSMA expression during culturing of 22Rv1 cells)."

Paragraph page 6: "Binding of the two bsAbs to PSMA and CD3 expressing cells (LNCaP or 22Rv1^{low} ; and Jurkat cells, respectively) was accessed by flow cytometry, which revealed EC₅₀ values of approximately 5nM and 9nM for PSMA binding and 10nM and 0.7nM for CD3 binding of the IgGsc and Fabsc molecule, respectively (Fig 2E-G)."

2. Fig 5 c - appears to be mislabeled - tumor eradication is seen at doses of 0.8, 1.0 and 2.0 but not at doses of 1.6, or 1.2 which is both internally inconsistent and contrary to the text in manuscript.

We sincerely apologize for mislabeling the figure 5c (old figure numbering) and have corrected this error (now presented as Fig 6C). Indeed, the tumor eradication was measured at doses of at 2µg, 1.6µg and 1.2µg (but not at a 1µg dose).

3. Table 1 is very difficult to read and understand. Perhaps an H-score to summarize both intensity and % cells + would have been more useful

*We thank the reviewer for this comment and have now replaced table 1 by a graph included in the figure 1D. Briefly, Values correspond to the number of sample with absent, weak, intermediate or high reactivity defined as containing no, approx. 1-10%, 11-50% and >50% specifically stained cells, respectively. The staining intensities (0, 1, 2 or 3) were determined for each sample and the H-score calculated include the sum of individual H-scores for each intensity level seen. The H-scores, ranging from 0 to 300, represent the SD calculated for each sample. Statistical analysis, comparing J591 to 10B3 intensity staining was performed using the unpaired t test (ns, non-significant; *p < 0.05, **p < 0.01 ***p < 0.001). The introduction of the H-score clearly shows the superiority of 10B3 in staining prostate carcinoma vessels and lung SSC tumor cells and this has largely facilitated the illustration of this finding.*

Referee #3 (Comments on Novelty/Model System for Author):

Referee #3 (Remarks for Author):

Zekri et al identified and engineered a bispecific antibody for the treatment of PSMA(+) cancer with T-cells. Patients with prostate cancer have limited treatment options and any novel approaches that limit toxicity while shrinking tumor are likely to have a major impact the field. In this manuscript, Zekri et al cover two important points: First they describe how this anti-PSMA clone differs from the conventionally used J591 clone, by showing how the epitope of 10B3 is more widely displayed in neovasculature of multiple cancer types. Second they engineer two T-cell bispecific antibodies using this sequence and evaluate their function in vivo, both pre-clinically and in three patients, providing first in human data for this T-cell bispecific antibody.

Although two BsAb formats were compared, it is unclear why either was chosen among the big list of possible antibody platforms.

We certainly agree with the reviewer that there are numerous different formats used for the construction of bispecific antibodies. It was not our intention to extensively compare a larger number of these. Rather, the “philosophy” of our group was and still is to construct reagents with a preserved affinity towards a target antigen and to avoid aggregation due to the use of single chains. With this guiding principle we have chosen for this paper to compare a representative “small” molecule with a rather short half-life and rapid elimination with a large molecule exhibiting relatively long serum half-life. With this we intended to contribute to an unresolved controversy as to the role of these parameters for in vivo antitumor efficiency. While it has been argued that small molecules may be superior for tumor localization due to better penetration, our data clearly demonstrate superior tumor uptake of the large bispecific molecule.

The following changes are recommended to more clearly articulate the functional relevance of 10B3 compared to J591, in vitro:

1) Fc-attenuation is mentioned but not clearly cited or spelled out in the manuscript. Which mutations were used to ablate Fc binding? How did this attenuation affect antigen binding, pharmacokinetics, and T cell homing?

The mutations used to attenuate the Fc part are noted in the material and methods section. That they result in a complete abrogation of Fc function is suggested by the data demonstrated in Fig. 4, as discussed in the paper. To additionally address the reviewers concerns concerning the effect of the mutations on pharmacokinetic, we compared the half-

life of CC-1 molecule with a mutated Fc-part to that, of an otherwise identical molecule with a wild type Fc part. 20µg of either molecule were injected into C57BL/6 mice and the serum concentration of the protein was determined using a bioluminescent cell-based assay system developed by Promega (Madison, USA). The results are presented below.

Figure legend: Comparison of serum half-lives of CC-1, equipped with an attenuated Fc part (Fcko), and an identical molecule carrying a wild type Fc part. 20µg of either molecule were injected into C57BL/6 mice and the serum concentrations were measured at the indicated time points using the Promega assay described in the methods section. Mean values and standard deviations obtained from groups of four animals per time point are indicated.

2) Reformat Table 1 to display the relative fraction of samples that stain at each intensity level. This can be both in a table, or even a graph, such that the differences between tumor types can be more clearly seen. In addition, statistical analysis between relevant groups should be presented to prove the novel reactivity with neovasculature.

We thank the reviewer for this valuable comment, which was also raised by reviewer1. Briefly, we replaced t table 1 by graph introducing an H-score and corresponding statistical analysis. We refer to our respective reply to comment “3 - minor criticisms” of reviewer 1 for details.

3) Also reformat Table 1 such that the frequency of staining within each intensity level can more clearly compared between J591 and 10B3. For example, splitting each tumor type into a 2-cell wide by 4-cell long table for each tumor type and antigen type (Tumor, Vasculature) would allow one to compare J591 and 10B3 side by side for each intensity level. A similar change can be made for the aggregate staining scores to allow for statistical comparison.

As mentioned above we have now replaced table 1 by a graph included in the figure 1D. We hope that this addresses the reviewer concerns.

4) It is not clear if these scores and the slides they were based on were evaluated by a trained pathologist. Although a pathologist is included in the author list. If he or she has evaluated these slides, it would be important to highlight this in the methods or even the text to strengthen the claims.

In fact, all slides were evaluated by the trained pathologist listed as an author (BS). The final graphs and figures of the revised manuscript were again controlled and approved by this author. A respective remark is now added in the material and methods section.

5) Figure EV3 shows very little difference between 10B3 and J591, relative to the slides shown in figure 1. Please comment on how representative these slides are of their respective staining intensity as displayed in table 1, and if necessary, provide additional examples of when these antibodies stain similarly or differently.

We thank the reviewer for addressing this point which was also raised by reviewer 1. To address it, we have conducted several immunohistochemistry experiments using Lung SSC tumor slices. The results are now presented in a new figure EV2, replacing the old figure EV3. Briefly, the data support the following conclusion:

- i- 10B3 stains SSC lung tumors better than J591(EV2A)*
- ii- 10B3 and CC-1 are staining tumor vessels (EV2A-C)*
- iii- 10B3 staining appears to be PSMA specific (EV2D)*

For more information and semi-quantitative analysis of immunohistological data, we refer to our reply to comment "1 and 2" of reviewer 1.

6) Figure EV4B does not clearly show the experimental intent of the authors. More robust quantification or single color controls or control tissues would be helpful.

We thank the reviewer for this valuable remark. The aim of the original figure was to demonstrate a simultaneous staining by the PSMA antibody 10B3 and an antibody directed to EpCAM. on lung SCC tumor cells, thereby demonstrating membrane PSMA expression. We have now provided the single panel colors, 10B3+dapi, EpCAM+dapi and the overlay picture (figure EV3B, upper panel). Additionally, we have performed an immunofluorescence

analysis with a PSMA negative lung SSC sample. Here it can be seen a specific staining of 10B3 on the vessels and no staining in the tumor cells although EpCAM positive. All results are included in the figure EV3B (lower panel).

7) Figure 4B needs to have the legend updated to address more clearly what 1 and 2 are labeling.

We thank the reviewer for pointing out this lack of clarity and have stated now in the figure legend that (1) represents the DU145 “PSMA negative” tumor and (2) the 22Rv1 “PSMA positive” tumor.

The following changes are recommended more clearly explain the current in vitro and in vivo BsAb data, and more robustly evaluate its potential in other indications:

1) The figure legend of Figure EV5 C-E seems to be inconsistent with the text. In the figure, CD3 binding is clearly superior for the IgGsc, while PSMA binding is moderately improved with the Fabsc, in contrast to the text, which describes it in the opposite way. Please either correct or clarify. This is also seen on page EV7, which shows the same results as EV5.

Indeed, the Figure EV5E (old figure numbering in the submitted manuscript) and Fig 2F (new figure numbering in the revised version) clearly demonstrates that binding of the Fabsc- to CD3 expressed on Jurkat cells is superior to that of the IgGsc molecule. This refers to EC_{50} values, not to the plateau level of staining. The difference in the plateau level can be explained by the different binding of the fluorescence labeled detection antibody to both formats. The superior binding of the Fabsc molecule is discussed in the paper. To clearly determine the affinities of the two molecules, we have performed additional Biacore measurements that confirmed the high affinity of the Fabsc molecule (Fig 2J).

2) Additionally, it would be useful to evaluate the CD3 binding to primary human PBMCs or T-cells, in addition to or instead of Jurkat cells.

We followed this suggestion and have performed additional FACS titrations on CD4 and CD8 T cells using PBMCs from three different healthy donors. The results are now included in the Fig 2H, I and show EC_{50} values similar to the ones obtained with Jurkat cells.

3) The results of Figure EV5C-E are somewhat surprising either way. Could this be from the staining method used? The anti-Fab2 may bind differently to the Fabsc compared to the IgGsc. For example, could the IgG-sc molecule be bound by two different secondary molecules? Additional controls to evaluate this would be valuable. Additionally, while gel-filtration data looks clear, numerical summary of each peak for the SEC-HPLC would be helpful. If these values differ substantially it may be worth re-testing the staining using purer preparations, i.e. without aggregates.

We agree with the referee that the different plateau levels observed in the FACS based binding experiments appears—at first glance- surprising but can be reconciled by the following assumption: binding of the large IgGsc molecule to CD3 is univalent (in accordance with the unexpectedly low affinity of this molecule). This results in an identical number of bound molecules at saturating concentrations. Binding of the detection antibody to the larger molecule is increased correspondingly. In contrast binding of the large IgGsc molecule to PSMA positive cells occurs in a bivalent manner, resulting in a reduction of bound molecules by a factor of 2 partly compensating for reduced molecular weight of the small Fabsc molecule.

With the respect to SEC- HPLC data, we have included now the numerical summary of each peak presented in a small graph in figure 2C. Additionally, it should be noted that all the experiments performed in this manuscript were done with aggregate free proteins.

4) Why does Fabsc show consistently better IFN and IL-2 release in vitro? This suggests that Fabsc may be aggregating in vitro, or may be contaminated in some way (endotoxin, etc). Please provide data supporting that Fabsc and IgG-sc have similar in vitro stabilities and endotoxin levels. Assuming they are the same, please provide a better explanation of why the Fab sc seems to perform so well. This would also impact why Fab-sc functions independent of tumor.

We thank the referee for this comment. In accordance with the data presented we hypothesis that:

i- The somewhat increased activity is due to increased CD3 affinity (as discussed above).

ii- With the respect to endotoxin level, only endotoxin-free (<0.5 EU/ml) molecules, in the monomer form were used in our experiments (see material and methods section).

iii- To further evaluate the stability of the molecules, we have performed a thermal shift assay experiment to address the stability of both proteins. The results presented as a first derivative in the figure 2D show similar stability with a melting temperature of 74°C and 73°C for the IgGsc- and Fabsc-molecules, respectively.

5) The binding data is referred to as affinity, which is slightly misleading. Given that PSMA binding kinetics were evaluated by SPR, it would be very helpful to both present this data earlier in the manuscript, (along with the other biochemical characterizations) and include the fitted affinity quantitation's. This should be done for both BsAb's and using CD3 as well. Doing so will allow for more appropriate conclusions to be made about binding affinities and binding kinetics (for comparison between different formats and antibody clones).

We thank the reviewer for his comment and apologize for not including the affinity quantitation in the submitted version of the manuscript. SPR graphs and the corresponding KD values are now included as a part of the figure EV1F (earlier in the manuscript, as suggested by the referee). The results demonstrate that the affinities of the chimeric 10B3 vs. chimeric J591 are comparable, in accordance with FACS titration data performed on PSMA positive cells (Fig EV1E).

To further answer the referees question regarding the binding affinities and kinetics of IgGsc- vs. Fabsc-molecules to the CD3 protein, we performed an SPR experiment using a recombinant CD3delta-epsilon heterodimer. The results included in figure 2J, confirm the considerably higher affinity of the Fabsc compared to CD3 binding relative to IgGsc.

6) The LNCAP data appears compelling, but it would be helpful to compare this against a J591 BsAb as well. Although this is not entirely necessary in the case of LNCAP, it would be useful to show that 10B3 can indeed control or shrink non-prostate PSMA(+) tumors in vivo. It is understood that this level of biology may not be possible to model using a cell line or PDX xenograft system, but an explanation of such limits would be helpful too.

With regard to this comment, we have to state that developing a new model (that is to show that non prostate PSMA tumors are shrunk by CC-1) is challenging. An established mouse model using non-prostate PSMA positive tumor is to our knowledge difficult, because so far we didn't found any non-prostate tumor cell line that reliably express PSMA. A PDX xenograft model would be interesting to establish because usually the tumor resemble the original tumors in patients, both histologically and genetically. For the time being, we didn't

develop such challenging model, at least for solid tumors, but definitely it will be something to do, despite the high cost of establishment of such models that must be overcome.

In the absence of in vivo data, a clearer explanation of why the authors believed 10B3 to be superior to J591 would be helpful. For example, does the reduced specificity of 10B3 warrant concern that 10B3 would be more toxic to normal vasculature? Could it also reduce the tumor targeting of the antibody, and therefore reduce T-cell infiltration?

We thank the reviewer for this valuable comment. With respect to vascular expression there is not much difference between the two PSMA antibodies in non-prostate cancer. Moreover, we have previously performed an extensive immunohistochemistry analysis using FDA approved array of normal human tissues and the results are part of the documents submitted to regulatory authority for approval of a clinical study with CC-1. Indeed, we didn't notice any significant PSMA staining on normal vascular cells in accordance with the literature (Chang et al, 1999).

7) More details are needed in the method section regarding the outline of the animal models used. Routes of administration, timing and doses need to be more clearly spelled out. In addition, justification is needed for starting treatment so soon after tumor implantation (24hr, page 8). Please also update the figure axis labels to distinguish days since treatment or days since implantation.

We provide now additional details in the material and methods with the respect of the mouse models used. The figure (6B, C) axis label is also updated. Regarding the metastasis mouse model, it is stated that the treatment starts soon after tumor injection, because the tumor cells are directly injected i.v in the blood circulation and reaches the primary site quickly (in our case the lung).

8) On page 9, the following sentence is unclear: "In our hands, this antibody, in marked contrast to steroids, interferes neither with T cell activation and tumor cell killing in vitro nor with antitumor activity in vivo. Please rephrase or clarify.

With thank the referee for raising this lack of clarity. In a recently accepted manuscript from our group (kauer et al.2020, in press), we demonstrated that Tocilizumab application, in contrast to steroids, does not interfere with the anti-tumor activity of CC-1 antibody. This sentence will be rephrased in the revised manuscript as follows:

“ In contrast to i.v. application of steroids that are widely used to prevent CRS, Tocilizumab does interfere with T cell activation neither in vitro nor in vivo (Kauer et al. 2020).”

9) It was unclear how the clinical trial was designed. The exact regulatory authority who approved the study (NCT 04104607) should be spelt out. For example, what was the justification for the dosing schedule and regimen. Additional, clearer quantitation of the CC-1 doses in Figure 6 would be helpful. In vitro cytokine release started at 0.01 nM (~2 ng/ml), peaking at 1 nM (~200 ng/ml) of BsAb. Serum concentrations of 200-500 ng/ml should hit the peak of cytokine storm - a dosing regimen seemingly incompatible with safe design in a phase I bispecific antibody trial. A serum level of 200-500 ng/ml for any bispecific antibody (whether BiTE or IgG formats) is very high and potentially lethal. It is possible that this construct was not very effective in activating T cells. The mention of steroids affecting anti-tumor activity should be referenced not as unpublished results but with citation from previous publications.

The federal regulatory authority in charge is the Paul-Ehrlich-Institute (PEI). Considerations of starting and target dose for the study are extensively elaborated on in the regulatory documents for this study (investigational medicinal product dossier, IMPD and investigators brochure (IB). For reasons of brevity only the starting and the target dose are now mentioned in the present paper.

We agree with the reviewer that the serum concentrations are high but respectfully maintain that they may be required to achieve optimal therapeutic activity. The reported results imply that such concentrations might be tolerated if Tocilizumab is applied as concomitant medication for prevention of CRS. We also would like to point out that doses up to 80mg have been safely applied in clinical studies with a bispecific CD20xCD3 antibody (Bannerji Ret al.(2019) Clinical Activity of REGN1979, a Bispecific Human, Anti-CD20 x Anti-CD3 Antibody, in Patients with Relapsed/Refractory (R/R) B-Cell Non-Hodgkin Lymphoma (B-NHL). Blood. <https://doi.org/10.1182/blood-2019-122451>).

We do not fully comply with the reviewer’s statement that T cell activation has been not very effective in these patients. Our data demonstrate that considerable activation of T cells in the peripheral blood occurred, however we concur that a more sustained activation might be desirable.

10) Regarding patient 2's response to tocilizumab, it is explained that they may have had a pre-existing anti-human immune response. If so, shouldn't this have also impacted the level of CC-1 in the blood, not just Toci?

We fully agree with the comment of the reviewer that for the patient 2, the pre-existing human anti-human neutralizing antibodies could have affected the CC-1 and Tocilizumab levels in the blood. This remark is now explicitly added to the revised version of the manuscript.

Again the authors wish to thank the reviewers for their insightful and helpful comments, which in our view have largely improved our manuscript and hope that the revised version is found acceptable for publication in EMBO Molecular Medicine.

For the authors,

G. Jung

6th Jul 2020

Dear Prof. Jung,

Thank you for submitting your revised manuscript to EMBO Molecular Medicine. Please accept my apologies for the unusual delay in getting back to you, which is due to the fact that I was expecting the report from referee #1, who was a critical referee during the first round of review. Despite several chasers and promises to provide a report, we still have not heard back from this referee, and thus prefer to make a decision now in order to avoid delaying the process further. Referee #3 kindly provided a report on your responses to both referee #1 and #3's comments that you will find attached below.

As you will see, referee #3 acknowledges your efforts to address the initial concerns, and recognizes that the manuscript has significantly improved. However, this referee also mentions issues that remain unanswered regarding both referees' initial reports, and additional experiments will be necessary to support the claims.

As EMBO Press encourages a single round of revisions only, we would normally reject the manuscript at this stage. However, as the reviewer recognizes (as we do) the potential clinical impact of the study and its interest for the community, we would like to exceptionally invite a second round of revisions. Please be aware that this will be the last chance for you to address the points raised by the referees. Particular attention should additionally be given to improve the flow and clarity of the manuscript.

When submitting your revised manuscript, please carefully review the instructions that follow below. Failure to include requested items will delay the evaluation of your revision:

- 1) A .docx formatted version of the manuscript text (including legends for main figures, EV figures and tables). Please make sure that the changes are highlighted to be clearly visible.
- 2) Individual production quality figure files as .eps, .tif, .jpg (one file per figure).
- 3) A .docx formatted letter INCLUDING the reviewers' reports and your detailed point-by-point responses to their comments. As part of the EMBO Press transparent editorial process, the point-by-point response is part of the Review Process File (RPF), which will be published alongside your paper.
- 4) Please provide up to 5 keywords.
- 5) Please reformat the references so as to have 10 authors listed before et al.
- 6) Please remove "Data not shown". As per our guidelines on "Unpublished Data" the journal does not permit citation of "Data not shown". All data referred to in the paper should be displayed in the main or Expanded View figures (see also referee's comments).
- 7) Please note that references for Fig. 1B and for panels A and B of Fig. 4 are missing in the main

manuscript text.

8) We would also encourage you to include the source data for figure panels that show essential data. Numerical data should be provided as individual .xls or .csv files (including a tab describing the data). For blots or microscopy, uncropped images should be submitted (using a zip archive if multiple images need to be supplied for one panel). Additional information on source data and instruction on how to label the files are available at

9) Thank you for providing "The paper explained" section. Please insert it in the main manuscript file.

10) For more information: There is space at the end of each article to list relevant web links for further consultation by our readers. Could you identify some relevant ones and provide such information as well? Some examples are patient associations, relevant databases, OMIM/proteins/genes links, author's websites, etc...

11) Author contributions: please use initials instead of full names. Most of the contributing authors have not been entered in the submission system. This has to be done before the paper can be accepted.

12) Thank you for providing a synopsis image. We could unfortunately not open the .jpeg file, and the pdf was too small. Could you please double-check and provide a file (png or jpeg) 550 px-wide x 400-px high?

13) As part of the EMBO Publications transparent editorial process initiative (see our Editorial at <http://embomolmed.embopress.org/content/2/9/329>), EMBO Molecular Medicine will publish online a Review Process File (RPF) to accompany accepted manuscripts.

In the event of acceptance, this file will be published in conjunction with your paper and will include the anonymous referee reports, your point-by-point response and all pertinent correspondence relating to the manuscript. Let us know whether you agree with the publication of the RPF and as here, if you want to remove or not any figures from it prior to publication.

I look forward to seeing a revised form of your manuscript as soon as possible.

Yours sincerely,

Lise Roth

Lise Roth, Ph.D
Editor
EMBO Molecular Medicine

To submit your manuscript, please follow this link:

Link Not Available

Photos 400-800 DPI

*Additional important information regarding figures and illustrations can be found at <http://bit.ly/EMBOPressFigurePreparationGuideline>

***** Reviewer's comments *****

Referee #3 (Remarks for Author):

The authors have made substantial effort to address the concerns of the reviewers.

Critique:

Referee #1:

The IHC of blood vessel comparing anti-CD31, 10B3 and J591 was helpful, although the resolution was too low to delineate the vessels from the stroma. A much higher magnification or a spot view is necessary to make the point. To the untrained eye, the vascular staining between J591 and 10B3 was near identical (contrary to the statement that J591 was less). Based on these stainings, hard to understand figure 1C and figure 1D, where there was difference in staining of SSC, and evidence of staining of vessels in prostate CA. This raises the possibility that 10B3 was recognizing an epitope shared by proteins distinct from PSMA. The inhibition study with soluble PSMA should be done on tumor sections from the same SCC tumor. However, even if there was inhibition, the potential for a cross reactive epitope on proteins other than PSMA could not be ruled out. If 10B3 was specific for PSMA, the discordance between prostate (where J591=10B3), and lung cancer (where 10B3>J591) was hard to explain. This was especially troubling with 10B3>J591 for Ve staining in these same prostate tumors.

Looking back at the BIACORE in figure EV1, 10B3 had substantially lower RU values (4-fold) than J591, inconsistent with the stronger staining of the SSC tumors if this binding was only to PSMA in figure EV2A or fig 1C, again raising the possibility that 10B3 was reacting with an epitope shared

between PSMA and other proteins.

The IHC in EV2C indicated staining, but again at such low magnification, its hard to make out details. The pattern of anti-CD31 also did not resemble those of 10B3 and should be stated clearly. The PSMA specific tracer should be disclosed. Was it PSMA-617? If it was, was it done with IRB approval or as part of standard clinical care using an approved agent?

The authors provided the preceding PSA levels, 30 days, 120 days, and 250 days prior to their high level on the day of treat, with the implication that the patients were not on any therapy for metastatic prostate cancer during those periods. Otherwise as standard of care, they would have had PSA levels. The authors should state that clearly in the description. The rapid drop in PSA was followed by a rapid rise within 20-30 days. The authors should also make a note of it.

In Fig1D, the use of unpaired T test requires the variance to be same between groups. Was that assumption valid?

Referee #3

Regarding Fc attenuation using deletions E233P; L234V; L235A;ΔG236; D265G; A327Q; A330S (EU-index). The authors should explain why so many mutations for silencing Fc, when 3 of them should be more than enough. The number of mutations could potentially make the protein a lot more immunogenic, suggested by the authors in the treated patient. Fig 4 showed cytokine release in the absence of tumors: IgGsc had no cytokine release presumably because of lack of binding to PBMC. Surprising that the anti-CD3 scFv had none, not even a little bit of activation of T cells. A more definitive way to demonstrate Fc silencing is binding (ELISA or BIACORE) to FcR or C1q, or functional assays such as complement activation/cytotoxicity, or ADCC.

The authors stated that FACS titration on CD4 and CD8 T cells in PBMC showed EC50 values similar to those obtained with Jurkat cells. But there was a clear difference in the curves between PBMC and Jurkat. The proposed explanation that "The difference in the plateau level can be explained by the different binding of the fluorescence labeled detection antibody to both formats," does not make sense", since it was only found in Jurkat and not in PBMC.

The statement that "The results demonstrate that the affinities of the chimeric 10B3 vs. chimeric J591 are comparable, in accordance with FACS titration data performed on PSMA positive cells (Fig EV1E)" was not accurate. The RU values were very different.

Developing a non-prostate PSMA model to validate the claims for vascular reactivity is understandable. In the absence of data, the authors should not claim superiority of 10B3 versus J591, given the equivocal IHC data presented.

The timing of treatment, 24 hours after tumor implantation was clearly before tumor establishment and should be stated clearly instead of using the mislabeling term "established" tumor systems. This has significant implications for the interpretation of the data. The term tumor regression was also misleading since there was no measurable tumor to regress from. A more appropriate description is "it suppressed tumor growth".

The ability to achieve such a high serum level of BsAb is noteworthy since serum levels in clinical BsAb studies not easily measurable even at high doses. The absence of CRS was even more remarkable. They should make a note of it in the results and in the discussion.

***** Reviewer's comments *****

Referee#3 (Remarks for Author):

The authors have made substantial effort to address the concerns of the reviewers.

We thank the reviewer for acknowledging our efforts and we think that the helpful comments have largely improved our manuscript. We hope that by conducting the additional amendments requested in the second round of revision, the issues of the reviewers are fully addressed.

Critique:

Referee#1:

1. The IHC of blood vessel comparing anti-CD31, 10B3 and J591 was helpful, although the resolution was too low to delineate the vessels from the stroma. A much higher magnification or a spot view is necessary to make the point. To the untrained eye, the vascular staining between J591 and 10B3 was near identical (contrary to the statement that J591 was less). Based on these stainings, hard to understand figure 1C and figure 1D, where there was difference in staining of SSC, and evidence of staining of vessels in prostate CA. This raises the possibility that 10B3 was recognizing an epitope shared by proteins distinct from PSMA.

We apologize that the description of our results was obviously misleading in the previous version of the manuscript. We agree with the reviewer that in Fig EV2B, staining of vessels with J591 and 10B3 in the used lung-SCC sample appears near identical. Indeed, this is reflected by the results shown in figure 1D, right panel, which revealed no significant difference between 10B3 and J591 with regard to staining of vessels. The exemplary lung-SCC IHC pictures shown in Fig 1C were chosen to illustrate the difference in staining between 10B3 and J591 with regard to tumor cells; with regard to staining of vessels, we interpret binding of the two antibodies as comparable (and thus again in line with the results shown in Fig 1D, right panel, even if admittedly it may appear to the untrained eye that binding of 10B3 may also be more pronounced with regard to vessels).

Overall, we interpret the IHC data obtained upon comparison of the bench mark PSMA antibody J591 and our novel PSMA binder 10B3 as follows:

- 1. In prostate cancer, staining of tumor cells with J591 and 10B3 is comparable, whereas significantly more pronounced binding of 10B3 compared to J591 to the neovasculature was observed in this disease entity.*

2. In SCC samples from a variety of cancer entities other than lung cancer (Non-lung SCC, n=34, 28 head and neck cancers, 3 cervical/uterus cancers, 2 bladder cancers, 1 penis carcinoma), very low binding to tumor cells was observed with both antibodies, with 10B3 displaying slightly (but significantly) more pronounced staining. Of note, with regard to the neovasculature of these non-lung SCC samples, pronounced binding of both J591 and 10B3 was observed, with 10B3 again displaying slightly but significantly higher binding. This is in line with previously published results (Chang et al., 1999) that reported on substantial PSMA expression on tumor vessels in various cancer entities as revealed upon staining with other PSMA antibodies.
3. The most important difference was observed with regard to binding of 10B3 and J591 in SCC of the lung: here we observed pronounced and significantly higher binding of 10B3 to tumor cells when compared to J591, whereas binding to the neovasculature was rather similar with both antibodies.

This is in line with the results of the IHC stainings depicted in Fig EV2 that were included in the first round of revisions based on the reviewer rightful comments as outlined above. Note that we have included higher magnification and spot views in Fig EV2 to optimize interpretation of the results as requested by the reviewer.

Again, we apologize that the initial description of the respective results was confusing. We are convinced that this circumstance has lead the reviewer to consider the possibility that 10B3 recognizes an epitope/protein distinct from PSMA. While this possibility can of course never be fully excluded, in our view the immunoprecipitation data shown in Fig EV1A-C (in particular the results shown in Fig EV1C obtained with lung-SCC samples) combined with the blocking experiments shown in Fig EV2D (suggested by the reviewer upon the first revision where binding of 10B3 to lung-SCC samples is prevented by preincubation with recombinant PSMA) clearly show that 10B3 indeed recognizes PSMA.

To address this critical and important issue of the reviewer, besides including the higher magnifications/spot views shown in Fig EV2, we largely amended the manuscript to describe our results clearly as follows:

Results section, page 5, line 15:

“Next, 10B3 and J591 were compared by immunohistochemistry on cryosections of various normal and malignant human tissues. Some reactivity of both antibodies was noted with normal prostate epithelium, proximal tubules of the kidney, salivary glands and -to a variable extent- hepatocytes as well as epithelial cells of mammary glands and the gastrointestinal tract, the latter possibly being an artefact caused by excessive mucin production in such tissues. Although the benchmark PSMA antibody J591 and our novel PSMA binder 10B3

showed a similar binding affinity to PSMA *in vitro* (Fig EV1E-F), when we comparatively analyzed their binding in a variety of different solid tumor samples, the results shown in Fig 1B-D and EV2 revealed that (i) in prostate cancer, staining of tumor cells with J591 and 10B3 is comparable, whereas significantly more pronounced binding of 10B3 compared to J591 to the neovasculature was observed in this disease entity; (ii) in SCC samples from a variety of cancer entities other than lung cancer (Non-lung SCC, n=34, 28 head and neck cancers, 3 cervical/uterus cancers, 2 bladder cancers, 1 penis carcinoma), very low binding to tumor cells was observed with both antibodies. 10B3 displayed slightly but significantly more pronounced staining. Substantial binding of both, J591 and 10B3 to the neovasculature of these non-lung SCC samples was observed, with 10B3 again displaying slightly but significantly higher staining. This is in line with previously published results reporting on substantial PSMA expression on tumor vessels in various cancer entities (Chang et al, 1999); (iii) the most important difference with regard to binding of 10B3 and J591 could be documented in SCC of the lung: here we observed pronounced and significantly higher binding of 10B3 to tumor cells when compared to J591, whereas binding to the neovasculature was rather similar with both antibodies. Of note, vascular as well as tumor cell staining could be blocked by recombinant PSMA, thereby demonstrating specificity (Fig EV2D). While it cannot fully be excluded that 10B3 might recognize an epitope/protein distinct from PSMA, in our view the immunoprecipitation data shown in Fig EV1A-C (in particular the results shown in figure EV1C that were obtained with lung-SCC samples) combined with these blocking experiments clearly show that 10B3 indeed recognizes PSMA.”

In addition, for information of the reviewer, we also uploaded high resolution images of IHC slides to further address this issue. Please see the attached file: SourceDataForEV2A-D.

2. The inhibition study with soluble PSMA should be done on tumor sections from the same SCC tumor.

We fully agree with the notion of the reviewer, and indeed we have used consecutive 3µm sections of the very same tumor in this experiment. We apologize that this was not described clearly in the previous version of the manuscript; it is now stated clearly in a respective paragraph in the methods section and the figure legend. Accordingly, the manuscript has been amended as follows:

Method section, page 16, line 16:

“...In the blocking experiment presented in Fig EV2D, directly consecutive tumor sections were stained in the absence or presence of recombinant PSMA protein (50µg/ml).”

Figure legend EV2:

“(A-D) Directly consecutive 3-µm sections (always obtained from the same lung SCC sample for each antibody panel) were analyzed by immunohistochemistry as described in the methods section.”

3. However, even if there was inhibition, the potential for a cross reactive epitope on proteins other than PSMA could not be ruled out. If 10B3 was specific for PSMA, the discordance between prostate (where J591=10B3), and lung cancer (where 10B3>J591) was hard to explain. This was especially troubling with 10B3>J591 for Ve staining in these same prostate tumors.

As stated above in our reply to comment 1, we apologize for not clearly describing the results in the previous version of the manuscript, which in our view has caused confusion. Please refer to comment 1 above for a detailed explanation how the manuscript was amended to clarify this issue.

4. Looking back at the BIACORE in figure EV1, 10B3 had substantially lower RU values (4-fold) than J591, inconsistent with the stronger staining of the SSC tumors if this binding was only to PSMA in figure EV2A or fig 1C, again raising the possibility that 10B3 was reacting with an epitope shared between PSMA and other proteins.

Also in regard to this point of the reviewer, we are convinced that a misleading description of our results has led to his notion and apologize for the nuisance we have caused. The observation of the reviewer that 10B3 has a lower RU than J591 is correct. However, RU values have no bearing on the calculation of K_{on} and K_{off} rates and the resulting KD values. The KD values that were calculated based on the Biacore data depicted in Fig EV1F are in agreement with the EC50 values obtained in the flow cytometry binding experiments (Fig EV1E). Based on these observations and considerations, we conclude that the affinities of J591 and 10B3 are rather comparable.

In greater detail: RU values reflect the interaction of an analyte (in our case the His-tagged PSMA) in solution with its respective ligand immobilized on the sensor surface (in our case

either J591 or 10B3). The theoretical analytic binding capacity of the surface in RU is given by R_{max} where

$$R_{max} = \text{ligand level} * (M_{W\text{analyte}} / M_{W\text{ligand}}) * \text{binding stoichiometry}$$

Of note, similar amounts of ligand (J591 or 10B3) were immobilized on the protein A sensor surface in the loading phase, which was reflected by similar RU starting values. We feel that the observed difference thus rather is caused by the differing binding stoichiometry of the two antibodies which bind different epitopes of the PSMA molecule. Indeed, as shown in Fig 1A, 10B3 binds to a conformational epitope of PSMA protein and shows a slow association kinetic (reflected by a low K_{on}) if compared to J591. This is compensated by the lower K_{off} rate of 10B3 compared to that of J591, overall resulting in a comparable K_D value.

To address this issue of the reviewer, the manuscript was amended as follows:

Discussion section, page 12, line 1:

“Analysis of binding by both, surface plasmon resonance and flow cytometry binding assays showed comparable affinities for J591 and 10B3 (Fig EV1E-F). This indicates that the differences observed by immunohistology might be attributable to a better accessibility of the 10B3-epitope once the cells are organized within a tissue.”

5. The IHC in EV2C indicated staining, but again at such low magnification, it's hard to make out details. The pattern of anti-CD31 also did not resemble those of 10B3 and should be stated clearly.

We thank the reviewer for raising this issue and have, as already stated above, included spot views in Fig EV2C to better illustrate binding of the Fabsc and IgGsc constructs to tumor associated vasculature. We also agree that – in contrast to staining with monospecific 10B3 and J591 antibodies shown in Fig EV2B - the pattern of anti-CD31 does not completely resemble staining results obtained with the PSMAxCD3 constructs containing 10B3 as target binder. In our view, this can, among others, be attributed to the fact that the bispecific Fabsc and IgGsc constructs besides PSMA bind to CD3+ T cells within the tumor. In addition, use of the PSMAxCD3 constructs for staining results in lower intensity due to the necessity to utilize a different detection system (please refer to the methods section for details) compared to the staining for CD31. Thus, the staining pattern observed in Fig EV2C indeed does not fully resemble that shown in Fig EV2B for the above described reasons. To clarify this issue, we have amended the manuscript as follows:

Figure legend EV2C:

“C. Binding of anti-CD31 as well as biotinylated IgGsc and Fabsc-molecules. Note that staining intensity with bsAbs is lower compared to CD31 due to the necessity to utilize a different detection protocol, and bsAb may additionally bind to tumor-infiltrating T cells, resulting in differential staining patterns. Arrows point to vessels. Scale 20µm.”

In addition, for information of the reviewer, we also uploaded high resolution images of IHC slides to further address this issue. Please see the attached file: SourceDataForEV2A-D.

6. The PSMA specific tracer should be disclosed. Was it PSMA-617? If it was, was it done with IRB approval or as part of standard clinical care using an approved agent?

We apologize that we did not describe clearly enough which tracer was used for PET imaging in the previous version of the manuscripts. This information has now been included in the figure legend EV5. We used PSMA-1007 which is a 18F-labeled peptide tracer for PET imaging that specifically binds to PSMA. It is used according to §13.2B AMG (German drug law). PSMA-PET has become standard clinical care in Germany.

To provide this information, the figure legend EV5A has been amended as follows:

“...Image data were acquired 60 min after i.v. injection of [18F]-PSMA-1007 (250-325 MBq), a labeled peptide tracer for PET imaging that specifically binds to PSMA. It is used according to §13.2B AMG (German drug law) for PSMA-PET which has become standard clinical care in Germany.”

7. The authors provided the preceding PSA levels, 30 days, 120 days, and 250 days prior to their high level on the day of treat, with the implication that the patients were not on any therapy for metastatic prostate cancer during those periods. Otherwise as standard of care, they would have had PSA levels. The authors should state that clearly in the description. The rapid drop in PSA was followed by a rapid rise within 20-30 days. The authors should also make a note of it.

We apologize but we are not sure that we understand the issue of the reviewer correctly. The three patients were suffering from metastatic prostate carcinoma refractory to standard medical treatment regimes. This is reflected by the rise of PSA values prior to antibody therapy. While the patients had undergone treatment in the (depicted) time prior to CC-1 therapy, they had not received any disease specific treatment for at least 4 weeks. This information has now been included in the figure legend EV5.

To further address the rightful comment of the reviewer, we included a statement that PSA levels rose again within 20-30 days after cessation of CC-1 treatment.

Accordingly, the manuscript has been amended as follows:

Results section, page 10, line 21:

“In all patients, profound T-cell activation and a rapid and marked decline of PSA levels were observed, which rose again 20-30 days after cessation of CC-1 treatment (Fig 7, Fig EV5B).”

Figure legend EV5B:

“B. Long term PSA values monitored prior, during (highlighted in light red), and after CC-1 therapy. After documented failure of established treatment, patients were free of disease specific therapy for at least 4 weeks prior to application of CC-1.”

8. In Fig1D, the use of unpaired T test requires the variance to be same between groups. Was that assumption valid?

We thank the reviewer for raising this important issue. Since we are comparing J591 and 10B3 staining in serial sections from the same tumor sample, this analysis indeed comprises paired samples which further were found to be not normally distributed. In this situation with paired, nonparametric data the Wilcoxon test is appropriate. The critique of the reviewer thus is fully correct, and we have statistically reanalyzed our data by using the Wilcoxon test instead of the T test. This resulted in even higher statistical significance of our findings.

To reflect this change, the figure legend 1D was amended as follows:

*“Semi quantitative analysis of binding of the PSMA antibodies 10B3 and J591 to cryosections from different tumor entities. For definition of the H-score reflecting binding intensity, refer to the methods section. Statistical analysis was performed using the paired, non-parametric Wilcoxon test (ns, non-significant; *p < 0.05, **p < 0.01 ****p < 0.001).”*

Referee#3

1. Regarding Fc attenuation using deletions E233P; L234V; L235A;ΔG236; D265G; A327Q; A330S (EU-index). The authors should explain why so many mutations for silencing Fc, when 3 of them should be more than enough. The number of mutations could potentially make the protein a lot more immunogenic, suggested by the authors in the treated patient.

We agree with the referee that higher numbers of mutations may result in higher immunogenicity. However, all the respective mutations were included to ensure the abolishment of FcR binding, according to the work published by Armour et al, 2003; Sazinsky et al, 2008 and Wines et al, 2000. In our view, this is particularly critical with T cell recruiting bsAbs since even a residual FcR binding may result in undesirable T cell activation.

To address this rightful issue of the reviewer, the manuscript was amended as follows:

Results section, page 7, line 1:

“ A combination of several point mutations or deletions (Armour et al, 2003; Sazinsky et al, 2008; Wines et al, 2000) was employed to ensure abolishment of Fc receptor (FcR) binding, as this may result in undesired T cell activation.”

2. Fig 4 showed cytokine release in the absence of tumors: IgGsc had no cytokine release presumably because of lack of binding to PBMC. Surprising that the anti-CD3 scFv had none, not even a little bit of activation of T cells. A more definitive way to demonstrate Fc silencing is binding (ELISA or BIACORE) to FcR or C1q, or functional assays such as complement activation/cytotoxicity, or ADCC.

Assuming that the comment of the reviewer aims to provide data regarding the Fc-silencing in CC-1, we have performed a series of new ELISA experiments to determine the binding of recombinant human FcR proteins to our CC-1 molecule with a mutated Fc-part (FcKO) and a “sister molecule” with a wild type Fc part (FcWT). To this end, the indicated his-tagged FcR proteins were immobilized to plastic followed by addition of titrated amounts of CC-1 molecules. For FcRn, the experiment was performed at both acidic (pH 6.0) and neutral pH (pH 7.2). The FcWT CC-1 variant showed a strong binding to all tested receptors. In contrast, the FcKO CC-1 showed no binding to either receptor except FcRn.

These results have been included in the new Fig EV4A and are described in the manuscript as follows:

Results section, page 7, line 3:

“Fc-silencing in CC-1 was confirmed by measuring the binding capacity to recombinant human FcR proteins. In contrast to the corresponding bsAb containing a wild type Fc part, CC-1 did not bind to any FcR except FcRn (Fig EV4A).”

Methods section, page 19, line 4:

“ FcR binding analysis was conducted using ELISA by coating wells his-tagged Fc γ RI, Fc γ RIIb, Fc γ RIIa Fc γ RIIa or FcRn protein (R&D Systems, Minneapolis, MN, USA) Then, bsAbs were added to the plate at the indicated concentrations, and binding was visualized using an HRP-conjugated goat anti human-Fc antibody (Jackson ImmunoResearch, West Grove, PA, USA). Unless indicated, all experiments were performed at a neutral pH.”

Figure legend EV4A:

“ A. Binding of CC-1 (FcKO) and a variant containing a wild type Fc-part (FcWT) to the indicated his-tagged FcR was determined by ELISA. All experiments were performed at a neutral pH except for FcRn, where binding was also evaluated at a pH of 6. Means of duplicate measurements are shown.”

3. The authors stated that FACS titration on CD4 and CD8 T cells in PBMC showed EC50 values similar to those obtained with Jurkat cells. But there was a clear difference in the curves between PBMC and Jurkat. The proposed explanation that "The difference in the plateau level can be explained by the different binding of the fluorescence labeled detection antibody to both formats," does not make sense", since it was only found in Jurkat and not in PBMC.

We thank the reviewer for this comment. In fact, as depicted in Figs. 2G,H,I, the statement that EC50 values for binding to CD3 are comparable between Jurkat and CD4+/CD8+ T cells holds true only for the IgGsc molecule, whereas the Fabsc has a lower EC50 activity –and a lower plateau level- with Jurkat cells. This is now stated clearly in the revised version of the manuscript. A potential explanation for this result could be a different architecture of the TCR/CD3 complex in the membrane of the two cell types, making the TCR/CD3 complex better accessible to univalent CD3 binders in case of Jurkat cells, resulting in a somewhat higher affinity in the latter.

The manuscript, Results section, page 7, line 12, has been now amended as follows:

“Binding of the two bsAbs to PSMA and CD3 expressing cells (LNCaP or 22Rv1low and Jurkat cells, respectively) was assessed by flow cytometry, which revealed EC50 values of approximately 5nM and 9nM for PSMA binding (LNCaP cells) and 10nM and 0.7nM for CD3 binding (Jurkat cells) of the IgGsc and Fabsc molecule, respectively (Figs 2E-G). Whereas the moderate loss of binding affinity of the N-terminal PSMA targeting part of the univalently binding Fabsc-molecule was expected, the lower CD3-affinity of the IgGsc-molecule that contains two CD3 binding single chain fragments was surprising. The latter was confirmed by measuring the CD3 binding affinities on CD4+ and CD8+ T cells using flow cytometry (Figs 2H, I) and kinetics by SPR measurements using recombinant CD3delta-epsilon (Fig 2J), respectively. EC50 values for binding to CD3 were comparable between Jurkat and CD4+/CD8+ T cells for the IgGsc molecule, whereas the Fabsc had a lower EC50 activity and a lower plateau level with Jurkat cells as compared to CD4+/CD8+ T cells. Altogether, these findings indicate that: (i) binding is moderately compromised by the bivalent C-terminal arrangement of the two single chains and (ii) the architecture of the TCR/CD3 complex in the membrane of Jurkat cells versus CD4+/CD8+ T cells may differ, with the latter being more accessible to univalent CD3 binders.”

The question why the plateau level with these cells is different for the Fabsc molecule is a separate issue, but may again be due a different accessibility, this time for the secondary detection antibodies used. Hypothetical as these explanations are, we stand with the data depicted in Fig 2 and appreciate that the reviewer pointed to their partly incorrect description regarding the binding of the Fabsc molecule to Jurkat cells.

4. The statement that "The results demonstrate that the affinities of the chimeric 10B3 vs. chimeric J591 are comparable, in accordance with FACS titration data performed on PSMA positive cells (Fig EV1E)" was not accurate. The RU values were very different.

We apologize for the confusion caused by the misleading description of our results. We have amended the respective paragraph in the manuscript and hope that the implemented changes are suited to sufficiently address the issue of the reviewer. As this issue has also been brought up by reviewer 1, we kindly ask to refer to our reply to the respective comment 4 of reviewer 1 above for details.

5. Developing a non-prostate PSMA model to validate the claims for vascular reactivity is understandable. In the absence of data, the authors should not claim superiority of 10B3 versus J591, given the equivocal IHC data presented.

As outlined in our reply to comment 1 of reviewer 1 above, we apologize that we did not describe the results of IHC analysis clearly enough in the previous version of the manuscript. We have now largely amended the description of the IHC data in the manuscript. We hope that the new description of these results and the amendments introduced in the manuscript are suited to sufficiently address this notion and rule out the doubts of the reviewer.

6. The timing of treatment, 24 hours after tumor implantation was clearly before tumor establishment and should be stated clearly instead of using the mislabeling term "established" tumor systems. This has significant implications for the interpretation of the data. The term tumor regression was also misleading since there was no measurable tumor to regress from. A more appropriate description is "it suppressed tumor growth".

We again apologize that obviously the description of our results was misleading and caused a misunderstanding. We have performed two different mouse models: (i) a metastasis mouse model (Fig 6A) and (ii) an established tumor mouse model (Fig 6B-C).

In the metastasis model, tumor cells were injected i.v. to allow for rapid trapping of cells within the microvasculature of the lung minutes after injection. In this model, treatment was started 24 hours after injection.

Notably, in the second "established tumor model", tumor cells were injected s.c. in the right flank of mice and treatment was conducted several days later when tumors had reached a diameter of 5mm.

To clarify and better described this issue; we have amended the manuscript as follows and hope that this can rule out the critique of the reviewer:

Results section, page 9, line 12:

"In a metastasis mouse model, mice were injected with LNCaP-cells i.v. (d0) followed by injection of PBMC (10^7) and antibodies (20 μ g) at d1 and d4. After 21 days, the numbers of metastatic cells in the lungs of the animals were determined by flow cytometry. In this model, the activity of the IgG molecule was clearly superior to that of the smaller molecule, although the Fabsc-reagent was injected repeatedly to compensate for its lower serum half-life (Fig 6A).

Next we employed a second tumor model, in which large tumors were established prior to treatment: LNCaP cells were injected into the right flank of the animals and treatment was started when tumors had reached a diameter of 5mm. Mice treated with the IgGsc-molecule

experienced a complete and long lasting tumor regression at the rather low dose of 2µg applied three times in weekly intervals together with human PBMC (Fig 6B)."

Discussion section, page 13, line 13:

"In both, a metastasis prevention model and in a second model where mice were bearing large established tumors, the IgGsc-molecule achieved a marked and prolonged antitumor effect, whereas the Fabsc-bsAb was clearly less efficient and ineffective, respectively."

7. The ability to achieve such a high serum level of BsAb is noteworthy since serum levels in clinical BsAb studies not easily measurable even at high doses. The absence of CRS was even more remarkable. They should make a note of it in the results and in the discussion.

Indeed, in our study, quite profound CC-1 concentrations were reached in patients and well tolerated, most likely because of the prophylactic application of Tocilizumab for CRS prevention. A remark on this issue is now added to the discussion in the revised version of the manuscript that reads as follows:

Discussion section, page 14, line 8:

"...Although it is obviously too early to draw definite conclusions, the achieved serum levels and the rapid and marked PSA reduction observed during treatment of these patients indicate that Tocilizumab may be suited to effectively attenuate the sequelae of cytokine release, thereby allowing for dosing of CC-1 that results in substantial serum levels."

As information for the reviewer, we also would like to add that the Promega assay used in our study allows the detection of CC-1 serum concentrations as low as 20ng/ml.

Again, we thank the reviewers for their insightful comments which in our view clarified several misleading points and hope this revised version is found acceptable for publication in EMBO Molecular Medicine.

For the authors,

G. Jung

20th Oct 2020

Dear Prof. Jung,

Thank you for the submission of your revised manuscript to EMBO Molecular Medicine. We have now received the enclosed report from referee #3 who is supportive of publication pending minor revisions (see below). I am thus pleased to inform you that we will be able to accept your manuscript pending the following final minor amendments:

1) Referee's comments:

We would like you to discuss the referee's points in writing. If you do have data at hand (clear example of prostate carcinoma vasculature and SPR data), we would be happy for you to include it, however we will not ask you to provide any additional experiments at this stage.

Please provide a letter INCLUDING my comments and the reviewer's reports and your detailed responses to their comments (as Word file).

2) Main manuscript text:

- Please answer/correct the changes suggested by our data editors in the main manuscript file (in track changes mode). This file will be sent to you in the next couple of days. Please use this file for any further modification.

- Please remove the yellow highlighted text.

- Abstract: please remove "rather" from the first sentence. We would also encourage you to rephrase the second sentence to make it clearer.

- Material and methods:

o Cells: Please indicate the origin of cells (human vs. murine), and whether they were authenticated (if applicable) and tested for mycoplasma contamination.

o Patients data: Please include a statement that informed consent was obtained from all subjects and the full statement that the experiments conformed to the principles set out in the WMA Declaration of Helsinki and the Department of Health and Human Services Belmont Report. (This also applies to patients' samples)

- Please include a Data availability section: Primary datasets produced in this study need to be deposited in an appropriate public database (see <https://www.embopress.org/page/journal/17574684/authorguide#dataavailability>). If not applicable, the following sentence should be included: "This study includes no data deposited in external repositories".

- Statistics: Please indicate in the legends or in the figures the exact $n=$ and $p=$ values, not a range, along with the statistical test used. Some people found that to keep the figures clear, providing a supplemental table with all exact p -values was preferable. You are welcome to do this if you want to.

- Please note that we now mandate that all corresponding authors list an ORCID digital identifier (missing for Latifa Zekri).

3) Source Data:

Thank you for providing Source Data. Please double-check the labelling of the pictures for Fig. EV2A

4) For more information:

<https://www.embopress.org/doi/full/10.15252/emmm.201910874> should be listed as a reference, not as a weblink in the FMI section.

5) Thank you for providing a synopsis. I slightly modified the text to fit our style and format, please let me know if you agree with the following:

Insufficient penetration of immune cells and therapeutic antibodies into the tumor core is a major limitation in the immunotherapy field. This study reports the development of a novel bispecific antibody, named CC-1, for improved dual targeting of tumor- and vascular cells in PSMA positive tumors.

- A novel PSMA antibody (10B3) exhibiting enhanced reactivity with tumor- and vascular cells in samples from prostate carcinoma and squamous cell carcinoma of the lung was generated.
- Two different bispecific antibodies comprising 10B3 and anti-CD3 single chain in a Fabsc- and IgGsc-format were constructed and characterized.
- In vivo application of both bispecific antibodies revealed that only the IgGsc-molecule localized at a given tumor site, resulting in effective tumor cell destruction.
- A first-in-man application of the IgGsc-molecule, designated CC-1, in three patients with metastasized prostate carcinoma, demonstrated profound T cell activation and a rapid decline of elevated PSA levels.
- A first-in-man clinical study in patients with prostate carcinoma is currently ongoing (NCT04104607).

6) As part of the EMBO Publications transparent editorial process initiative (see our Editorial at <http://embomolmed.embopress.org/content/2/9/329>), EMBO Molecular Medicine will publish online a Review Process File (RPF) to accompany accepted manuscripts.

In the event of acceptance, this file will be published in conjunction with your paper and will include the anonymous referee reports, your point-by-point response and all pertinent correspondence relating to the manuscript. Let us know whether you agree with the publication of the RPF and as here, IF YOU WANT TO REMOVE OR NOT any figures from it prior to publication.

I look forward to receiving your revised manuscript.

Yours sincerely,

Lise Roth

Lise Roth, PhD
Editor
EMBO Molecular Medicine

To submit your manuscript, please follow this link:

Link Not Available

The system will prompt you to fill in your funding and payment information. This will allow Wiley to

send you a quote for the article processing charge (APC) in case of acceptance. This quote takes into account any reduction or fee waivers that you may be eligible for. Authors do not need to pay any fees before their manuscript is accepted and transferred to our publisher.

***** Reviewer's comments *****

Referee #3 (Remarks for Author):

Want to thank the authors for making substantial improvements in the manuscript. The following comments should be easily addressable.

1. It's still unclear which IHC images provide evidence that 10B3 stains prostate carcinoma vasculature better than J591. The text references Figure 1B and EV2 for this claim, but the image in 1B does not point out vasculature, and EV2 only shows slides from lung SSC. The authors should provide a clear example of the prostate carcinoma vasculature that contributed to the summarized results in 1D.

2. A few aspects of the SPR data in figure EV1F are worth addressing. For J591, the minor differences in RU between concentrations imply the chip is nearing saturation before switching to buffer. The SPR data for 10B3, however, shows a clear linear relationship between concentration and RU values, implying the chip is at an equilibrium instead of saturation prior to running buffer. Additionally, 10B3 seems to show less reduction in RU over time compared to J591. Considering the overall KD values are relatively similar, these observations imply different values in K_{on} and K_{off} between the two antibodies. It would be meaningful to show the K_{on} and K_{off} values of the two antibodies along with the KDs. Also, 10B3 showed such minimal loss in RU over 1000 seconds that the authors may not have adequately been able to determine the K_{off} . Performing SPR at 37°C or increasing the time for k_{off} measurement should allow for a more reliable measurement of these kinetic properties.

Manuscript number: EMM-2019-11902

***** Reviewer's comments *****

Referee #3 (Remarks for Author):

1. It's still unclear which IHC images provide evidence that 10B3 stains prostate carcinoma vasculature better than J591. The text references Figure 1B and EV2 for this claim, but the image in 1B does not point out vasculature, and EV2 only shows slides from lung SSC. The authors should provide a clear example of the prostate carcinoma vasculature that contributed to the summarized results in 1D.

To address this comment, we have now included an exemplary staining picture in the figure EV2A reflecting the superiority of 10B3 staining in prostate carcinoma vessels over J591. Accordingly, we have referenced to this new figure in the results section (page 5, line 26) and the figure EV2A legend was amended as follow:

“A. Directly consecutive 3- μ m sections obtained from prostate carcinoma samples were stained with 10B3 and J591 mAbs. Arrows point to vessels. Tu: tumor. HE: Hematoxylin/Eosin staining. Scale 30 μ m.”

2. A few aspects of the SPR data in figure EV1F are worth addressing. For J591, the minor differences in RU between concentrations imply the chip is nearing saturation before switching to buffer. The SPR data for 10B3, however, shows a clear linear relationship between concentration and RU values, implying the chip is at equilibrium instead of saturation prior to running buffer. Additionally, 10B3 seems to show less reduction in RU over time compared to J591. Considering the overall KD values are relatively similar, these observations imply different values in K_{on} and K_{off} between the two antibodies. It would be meaningful to show the K_{on} and K_{off} values of the two antibodies along with the KDs. Also, 10B3 showed such minimal loss in RU over 1000 seconds that the authors may not have adequately been able to determine the K_{off} . Performing SPR at 37°C or increasing the time for k_{off} measurement should allow for a more reliable measurement of these kinetic properties.

We appreciate the reviewer comments and we agree with his notion that the overall KD values are relatively similar and that these observations imply different values in K_{on} and K_{off} for the two antibodies. Accordingly, we provided an updated figure EV1F with an increased time for the dissociation phase (additional 200s) as requested by the referee. We have also amended the figure by including a new table comprising the corresponding K_a , K_d and KD values.

Manuscript number: EMM-2019-11902

We again thank the referee for his insightful comments and hope that his issues thereby are convincingly addressed.

For the authors,

G. Jung

1st Dec 2020

Dear Prof. Jung,

We are pleased to inform you that your manuscript is accepted for publication and is now being sent to our publisher to be included in the next available issue of EMBO Molecular Medicine!

Congratulations on your interesting work,

With my best wishes,

Lise

Lise Roth, Ph.D
Editor
EMBO Molecular Medicine

Follow us on Twitter @EmboMolMed
Sign up for eTOCs at embopress.org/alertsfeeds

*** ** IMPORTANT INFORMATION ** **

SPEED OF PUBLICATION

The journal aims for rapid publication of papers, using the advance online publication "Early View" to expedite the process: A properly copy-edited and formatted version will be published as "Early View" after the proofs have been corrected. Please help the Editors and publisher avoid delays by providing e-mail address(es), telephone and fax numbers at which author(s) can be contacted.

Should you be planning a Press Release on your article, please get in contact with embomolmed@wiley.com as early as possible, in order to coordinate publication and release dates.

LICENSE AND PAYMENT:

All articles published in EMBO Molecular Medicine are fully open access: immediately and freely available to read, download and share.

EMBO Molecular Medicine charges an article processing charge (APC) to cover the publication costs. You, as the corresponding author for this manuscript, should have already received a quote with the article processing fee separately. Please let us know in case this quote has not been received.

Once your article is at Wiley for editorial production you will receive an email from Wiley's Author

Services system, which will ask you to log in and will present you with the publication license form for completion. Within the same system the publication fee can be paid by credit card, an invoice, pro forma invoice or purchase order can be requested.

Payment of the publication charge and the signed Open Access Agreement form must be received before the article can be published online.

PROOFS

You will receive the proofs by e-mail approximately 2 weeks after all relevant files have been sent to our Production Office. Please return them within 48 hours and if there should be any problems, please contact the production office at embopressproduction@wiley.com.

Please inform us if there is likely to be any difficulty in reaching you at the above address at that time. Failure to meet our deadlines may result in a delay of publication.

All further communications concerning your paper proofs should quote reference number EMM-2019-11902-V4 and be directed to the production office at embopressproduction@wiley.com.

Thank you,

Lise Roth, Ph.D
Scientific Editor
EMBO Molecular Medicine

Corresponding Author Name: Gundram Jung
Journal Submitted to: EMBO Molecular Medicine
Manuscript Number: EMM-2019-11902